# Tailing and degradation of Argonaute-bound small RNAs protect the genome from uncontrolled RNAi

Paola Pisacane[1] & Mario Halic[1]

RNAi is a conserved mechanism in which small RNAs induce silencing of complementary targets. How Argonaute-bound small RNAs are targeted for degradation is not well understood. We show that the adenyl-transferase Cid14, a member of the TRAMP complex, and the uridyl-transferase Cid16 add non-templated nucleotides to Argonaute-bound small RNAs in fission yeast. The tailing of Argonaute-bound small RNAs recruits the 3′–5′ exo-nuclease Rrp6 to degrade small RNAs. Failure in degradation of Argonaute-bound small RNAs results in accumulation of 'noise' small RNAs on Argonaute and targeting of diverse euchromatic genes by RNAi. To protect themselves from uncontrolled RNAi, *cid14Δ* cells exploit the RNAi machinery and silence genes essential for RNAi itself, which is required for their viability. Our data indicate that surveillance of Argonaute-bound small RNAs by Cid14/Cid16 and the exosome protects the genome from uncontrolled RNAi and reveal a rapid RNAi-based adaptation to stress conditions.

[1] Department of Biochemistry, Gene Center, University of Munich, 81377 Munich, Germany. Correspondence and requests for materials should be addressed to M.H. (email: halic@genzentrum.lmu.de).

Small RNA (sRNA) silencing pathways are involved in the cellular control of gene expression and protection of the genome against mobile repetitive DNA sequences, retroelements and transposons[1–3]. The Argonaute family of proteins binds sRNAs that interact with target RNAs through base-pairing interactions. sRNAs promote DNA and chromatin modifications, translational inhibition and degradation of the complementary RNAs to induce gene silencing[1–3].

In fission yeast, sRNAs guide the RNA-induced transcriptional silencing (RITS) complex to centromeric repeats to induce histone 3 lysine 9 (H3K9) methylation and heterochromatin formation[3–6]. A subclass of short interfering RNAs (siRNAs) is generated independently of H3K9 methylation and heterochromatin protein 1 (HP1), indicating that heterochromatin is not a prerequisite for siRNA generation[7,8]. We previously described a distinct class of Dicer-independent sRNAs, called primal sRNAs (priRNAs)[8]. priRNAs are generated from single stranded RNAs by a trimming and protection mechanism that requires joint activity of Argonaute and the PARN family nuclease Triman (Tri1)[9]. Argonaute (Ago1), loaded with longer sRNA precursors, recruits Triman to generate mature priRNAs and siRNAs that are 22 nucleotides long. Longer sRNA precursors accumulating in *tri1Δ* cells are not competent in guiding Argonaute to slice complementary targets[9]. Our previous work showed that priRNA and siRNA trimming is required for *de novo* assembly of heterochromatin at centromeric repeats and *mat* locus and for maintenance of heterochromatin at developmental genes[9].

We also observed non-templated uridine(s) and adenine(s) at the 3′ end of Argonaute-bound sRNAs[8,9]. Addition of non-templated nucleotides to the 3′ end of sRNAs (RNA tailing) has been described in several organisms[8,10–13]. In plants uridylation of siRNAs and microRNAs (miRNAs) promotes their degradation[14,15], while adenylation of miRNAs was suggested to protect them from degradation[11]. In worms and mammals, the RNA-binding protein Lin28 recruits TUTases TUT4/7 to pre-*let-7*, resulting in pre-miRNA oligouridylation and degradation by the exonuclease Dis3L2 (refs 12,16,17). Defective pre-miRNAs that lack intact 3′ overhangs are also uridylated by TUT4/7 and targeted for destruction via the exosome[18]. Whispy adenylates maternally inherited miRNAs in flies and promotes their removal[19]. Although tailing and the resulting degradation of miRNA precursors has been described, it remains unclear how mature sRNAs can be actively removed from Argonaute and which nuclease(s) eliminate them.

In this study, we identified two non-canonical nucleotidyltransferases Cid14 and Cid16 that add non-templated nucleotides to the 3′ end of Argonaute-bound sRNAs and target them for elimination. Cid16 is a predicted poly(A) polymerase and has not been characterized yet[20,21]. Cid14 is a component of the nuclear TRAMP complex that provides target specificity to the exosome[22,23]. TRAMP oligo-adenylates unstable non-coding RNAs (ncRNAs) in *S. cerevisiae* and hands these RNAs to the Rrp6 or the core exosome[23–25]. The TRAMP complex and Rrp6 play also an essential role in the processing of rRNA, tRNAs and small nuclear and nucleolar RNAs[23,26]. Although the TRAMP complex is evolutionary conserved, the function of the mammalian and fission yeast TRAMP complex seems to be more restricted to rRNA biogenesis[27,28].

In *cid14Δ* cells centromeric transcripts show small accumulation that implicated Cid14 in RNAi-mediated heterochromatin formation in fission yeast[22,29]. We show here that Cid14 and Cid16 target Argonaute-bound sRNAs to the exosome and promote their degradation. We show *in vivo* and *in vitro* that Cid16 adds uridine(s) and Cid14 adenine(s) to the 3′-end of Argonaute-bound sRNAs. This addition of non-templated nucleotides to the Argonaute-bound sRNAs by Cid14 and Cid16 *in vitro* recruits the 3′–5′ exonuclease Rrp6 which actively removes and degrades sRNAs that are bound by Argonaute. The degradation of sRNAs on Argonaute is essential to reduce the 'noise' coming from the binding of various degradation products by Argonaute and ensures fidelity of sRNA-mediated silencing. In *cid14Δ* and *rrp6Δ*[9] cells, we observed that RNAi targets protein coding genes and rRNA, indicating that turnover of Argonaute-bound sRNAs protects the genome from spurious RNAi. Interestingly, in *cid14Δ* and *cid14Δcid16Δ* cells, RNAi targets Rdp1, an RNA-dependent RNA polymerase essential for siRNA generation[8,30]. This suppresses RNAi. Over-expression of Rdp1 in *cid14Δ* cells severely impairs their viability, showing that reduced levels of Rdp1 protect the genome from even more harmful, uncontrolled RNAi that appears in these cells. This is an example of rapid adaptation that cells acquired in response to stress conditions. Our work reveals a surveillance mechanism of Argonaute-bound sRNAs that comprises Cid16, the TRAMP complex and Rrp6. This surveillance mechanism reduces the levels of 'noise' sRNAs bound by Argonaute and increases fidelity of RNAi.

## Results

**Ago1-bound sRNAs are modified at the 3′ end.** We sequenced Argonaute-bound sRNAs and total sRNAs (20–30 nucleotides long RNAs) from wild-type cells. In the total sRNA fraction, centromeric siRNAs comprise only ∼25% of sRNAs and many reads are degradation products of rRNA, tRNAs and mRNAs (Fig. 1a). In the Argonaute-bound sample, sRNAs show strong preference for 5′ U and the majority of sRNAs are centromeric siRNAs[8] (Fig. 1a; Supplementary Fig. 1a). Sequencing of Argonaute-bound and total sRNAs revealed that centromeric siRNAs from all centromeric loci are loaded on Argonaute, including siRNAs that are generated at all boundary elements between heterochromatin and euchromatin (Fig. 1b). Although sRNAs target transposons in many organisms, we found only few Dicer-independent priRNAs and no siRNAs at the transposable element Tf2 and LTR elements. Our data show that Tf2 and LTR are not silenced by RNAi in fission yeast (Supplementary Fig. 1b,c).

We analysed the sequence of Argonaute-bound and total sRNAs and observed that >20% of Argonaute-bound sRNAs have 1–2 non-templated nucleotides at the 3′ end (Fig. 1c)[8]. More than 70% of those non-templated nucleotides at the 3′ end of sRNAs are adenines, while 25% are uridines (Fig. 1c). Overall, ∼25% of siRNAs and priRNAs generated from centromeric transcripts, mRNAs and ncRNAs are adenylated or uridylated at the 3′ end (Supplementary Fig. 1d). On the contrary, only ∼5% of sRNAs generated from tRNAs and sense rRNA had non-templated nucleotides at the 3′ end (Supplementary Fig. 1d). sRNAs generated from transcripts that are antisense to rRNA were adenylated more frequently, similarly to priRNAs generated from mRNA transcripts (Supplementary Fig. 1d). This suggests that sRNAs originating from centromeric region, mRNAs and ncRNAs are processed in the same way, while priRNAs generated from sense rRNA and tRNAs form a distinct class of sRNAs. Alternatively, priRNAs generated from tRNAs and sense rRNA might be loaded on Argonaute during cell disruption and are not genuine *in vivo* Argonaute-bound sRNAs.

In contrast to Argonaute-bound sRNAs, total sRNAs were adenylated or uridylated less frequently (Fig. 1c). Centromeric siRNAs were modified at the similar rate in both the total sRNA and the Argonaute-bound samples suggesting that majority of the siRNAs deriving from centromeric repeats are loaded onto Argonaute (Fig. 1d). sRNA degradation products originating from mRNAs, tRNAs and rRNA were rarely adenylated or

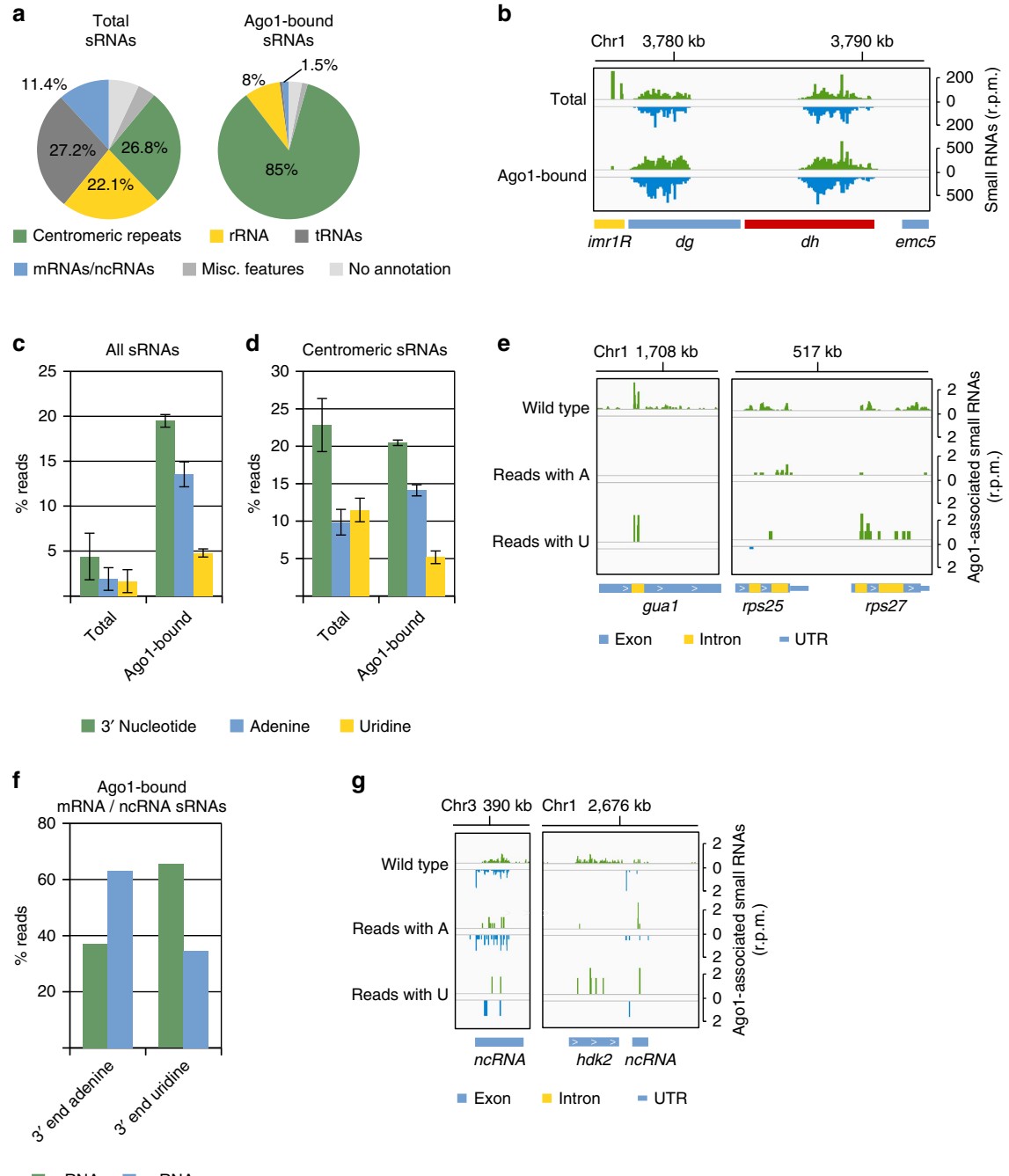

**Figure 1 | Argonaute-bound sRNAs have non-templated nucleotides at the 3' end.** (**a**) Argonaute-bound and total sRNAs were analysed by high-throughput sequencing from wild-type cells and classified as indicated below the pie charts. Pie charts illustrate percentages for the individual sRNA classes relative to the total number of reads for each strain. (**b**) Argonaute-bound and total sRNA reads from wild-type cells were plotted over centromeric region. The location of genes is indicated below the sRNA peaks. Reads from + and − strands are depicted in green and blue, respectively. Scale bars on the right denote sRNA read numbers normalized per one million reads. (**c**) Quantification of sRNAs that have non-templated nucleotides at the 3' end in Argonaute-bound and total sRNA sample. Error bars indicate s.e.m. of two independent sRNA-sequencing experiments. (**d**) Quantification of centromeric sRNAs that have non-templated nucleotides at the 3' end in Argonaute-bound and total sRNA sample. Error bars indicate s.e.m. of two independent sRNA-sequencing experiments. (**e**) Argonaute-bound sRNA reads from wild-type cells were plotted over euchromatic region. Reads having non-templated adenine(s) or uridine(s) at the 3' end are shown in separated tracks. The location of genes is indicated below the sRNA peaks. Reads from + and − strands are depicted in green and blue, respectively. Scale bars on the right denote sRNA reads numbers normalized per one million reads.
(**f**) Quantification of Argonaute-bound sRNAs that have non-templated adenine(s) or uridine(s) at the 3' end. Non-templated adenine is enriched at sRNAs that originate from non-coding and antisense transcripts. Non-templated uridine is enriched at sRNAs that originate from mRNAs. (**g**) Argonaute-bound sRNA reads from wild-type cells were plotted over euchromatic genes. Reads having non-templated adenine(s) or uridines(s) at the 3' end are shown in separated tracks. The location of genes is indicated below the sRNA peaks. Reads from + and − strands are depicted in green and blue, respectively. Scale bars on the right denote sRNA reads numbers normalized per one million reads.

uridylated in the total sRNA sample (Supplementary Fig. 1e). These data suggest that only Argonaute-bound sRNAs are adenylated or uridylated and suggest that non-templated nucleotides are added to the 3′ end of sRNAs after they are loaded onto Argonaute (Supplementary Fig. 1d,e).

We extracted sRNA reads that are adenylated or uridylated at the 3′ end. Both adenylation and uridylation of sRNAs are distributed over the whole genome (Fig. 1e; Supplementary Fig. 1f,g) with reduction at rRNA and tRNAs (Supplementary Fig. 1d). We observed that priRNAs generated from introns are more often adenylated or uridylated, indicating that intronic priRNAs are modified more frequently than other priRNAs (Fig. 1e). In addition, Argonaute-bound priRNAs generated from non-coding antisense RNAs are more frequently adenylated, while priRNAs generated from sense mRNA transcripts are more often uridylated (Fig. 1f,g). Although we observed some variation in addition of non-templated nucleotides at the 3′ end of different classes of sRNAs, our data show that all classes of Argonaute-bound sRNAs are adenylated and uridylated.

**Cid14 adenylates and Cid16 uridylates Ago1-bound sRNAs.** To determine the nucleotidyl-transferases that add non-templated nucleotides to the 3′ end of sRNAs, we deleted all non-canonical nucleotidyl-transferases in fission yeast. Deletion of Cid14 showed strong reduction in addition of adenine(s) to the 3′ end of Argonaute-bound sRNAs indicating that Cid14 is the main adenyl-transferase[8] (Fig. 2a; Supplementary Fig. 2a). In cid16Δ cells uridylation of the 3′ end of sRNAs was lost, showing that Cid16 is the only sRNA uridyl-transferase (Fig. 2a). In cid1Δ, cid11Δ, cid12Δ and cid13Δ cells, we observed only a minor reduction in addition of non-templated adenine(s) to the 3′ end of sRNAs (Supplementary Fig. 2b). We also generated double mutants of all nucleotidyl-transferases to find if some might be redundant. In cid14Δcid16Δ cells, we observed a strong reduction in adenylation and uridylation of Argonaute-bound sRNAs (Fig. 2a; Supplementary Fig. 2c). This shows that Cid14 and Cid16 are the main nucleotidyl-transferases that add non-templated nucleotides to sRNAs in fission yeast. In cid14Δcid12Δ, sRNA tailing was further reduced indicating that Cid14 and Cid12 might be partially redundant. All classes of Argonaute-bound sRNAs showed similar loss in adenylation and uridylation in cid14Δcid16Δ cells (Supplementary Fig. 2d). Consistent with deficiency in adenylation and uridylation, Argonaute-bound sRNAs in cid14Δcid16Δ cells are shorter than in wild-type cells (Supplementary Fig. 2e).

Although uridylation and adenylation of sRNAs were lost, we observed only a small reduction of centromeric dg and dh siRNAs in cid16Δ, cid14Δ and cid14Δcid16Δ cells compared to wild type and dcr1Δ cells (Fig. 2b,c). Only siRNAs generated at the boundary element IRC3 were strongly reduced in cid14Δ[31] and eliminated in cid16Δ cells. Biogenesis of siRNAs at IRC3 is affected in most mutants[8,9], and siRNA generation at this repeat is sensitive to any perturbation suggesting that this is likely an indirect effect. Centromeric siRNAs in the total RNA population were present near wild-type levels in both cid14Δ and cid16Δ cells as shown by northern blotting analysis (Supplementary Fig. 3a). We found, however, a reduction of centromeric siRNAs in the total RNA pool in cid14Δcid16Δ cells, which is consistent with the sequencing of Argonaute-bound sRNAs (Fig. 2b,c; Supplementary Fig. 3a). Centromeric dg transcripts were 2–3 fold upregulated and H3K9me2 at centromeric repeats was reduced 2–3 fold in these mutants (Fig. 2d; Supplementary Fig. 3b). Our data show that in cid14Δ, cid16Δ and cid14Δcid16Δ cells heterochromatic silencing at centromeric repeats is only moderately reduced when compared to dcr1Δ cells (Fig. 2d).

This indicates that addition of non-templated nucleotides to the 3′ end of sRNAs by Cid14 and Cid16 is not essential for silencing of centromeric repeats.

**Cid14 and Cid16 protect the genome from uncontrolled RNAi.** In cid14Δ and cid14Δcid16Δ cells, Argonaute was associated with a higher amount of sRNAs than in wild-type cells, suggesting that Cid14 promotes degradation of sRNAs (Fig. 3a). Some classes of sRNAs accumulated more than others in cid14Δ and cid14Δ cid16Δ cells. For example, Argonaute-bound priRNAs generated from mRNAs, ncRNAs and introns accumulated in cid14Δ and cid14Δcid16Δ cells, and to a lower extent in cid16Δ cells. (Fig. 3b; Supplementary Fig. 3c). This indicates that Cid14 and Cid16 are required for elimination of these subclasses of sRNAs.

We also observed the generation of new siRNAs at many euchromatic genes in cid14Δ and cid14Δcid16Δ cells and to a lower level in cid16Δ cells (Fig. 3c). In wild-type cells we could not detect any siRNAs at these genes (Fig. 3c). Initiation of siRNA generation[8,9] is consistent with the increase of Argonaute-bound priRNAs in cid14Δ and cid14Δcid16Δ cells (Fig. 3a,b). Accumulation of priRNAs will trigger generation of secondary siRNAs at euchromatic loci[9]. It was observed that in cid14Δ cells an increased number of sRNAs maps to rRNA[31]. In addition to rRNA, we observed that many Argonaute-bound siRNAs map to mRNAs and ncRNAs in cid14Δ and to an even greater extent in cid14Δcid16Δ cells (Fig. 3c; Supplementary Fig. 3d). siRNAs generated at ectopic loci are fully functional and reduce abundance of targeted RNAs in cid14Δ and cid14Δcid16Δ cells (Fig. 3d,e; Supplementary Fig. 3e). We did not detect H3K9 methylation at genes targeted by RNAi in cid14Δ cells indicating that RNAi does not establish heterochromatin at these loci (Supplementary Fig. 4a). These data also show that RNAi in fission yeast can be uncoupled from heterochromatin and that H3K9 methylation is not required for siRNA generation and silencing. We did find, however, an increase in H3K9 methylation at rDNA in cid14Δ cells and cid14Δcid16Δ (Supplementary Fig. 4b). Our data suggest that adenylation of sRNAs by Cid14 protects the genome from uncontrolled RNAi implicating that adenylation of Argonaute-bound sRNAs promotes their degradation. A defect in surveillance of Argonaute-bound sRNAs results in accumulation of priRNAs that will guide RNAi to unwanted targets.

**Rapid adaptation to stress conditions.** One of the genes, where a high amount of siRNAs was generated in cid14Δ and cid14Δ cid16Δ cells, was the RNA-dependent RNA polymerase rdp1 (Fig. 4a). In addition to rdp1, in cid14Δcid16Δ cells RNAi also targeted ago1 gene (Supplementary Fig. 4c). Rdp1 synthesizes dsRNA that is processed by Dicer into siRNAs[30,32]. Synthesis of dsRNA by Rdp1 is essential for the generation of centromeric siRNAs and heterochromatin formation at centromeric repeats[8,30]. The Rdp1 mRNA was 1.5–2 fold reduced in cid14Δ cells and 4 fold in cid14Δcid16Δ cells in qPCR and RNAseq experiments (Fig. 4b; Supplementary Fig. 3e). H3K9 methylation, however, was not increased at the rdp1 gene in cid14Δ and cid14Δcid16Δ cells indicating that RNAi silences rdp1 mRNA independently of heterochromatin (Supplementary Fig. 4d). Our data suggest that in cid14Δ and cid14Δcid16Δ cells, RNAi targets rdp1 mRNA to protect the genome from uncontrolled RNAi. Over-expression of rdp1 in cid14Δ cells strongly impaired growth of these cells indicating that reducing Rdp1 level is essential for viability of cid14Δ cells (Fig. 4c; Supplementary Fig. 4e). We sequenced Argonaute-bound sRNAs from cid14Δ cells over-expressing rdp1 and observed a strong increase of sRNAs generated from rRNA (Fig. 4d). H3K9me was increased at

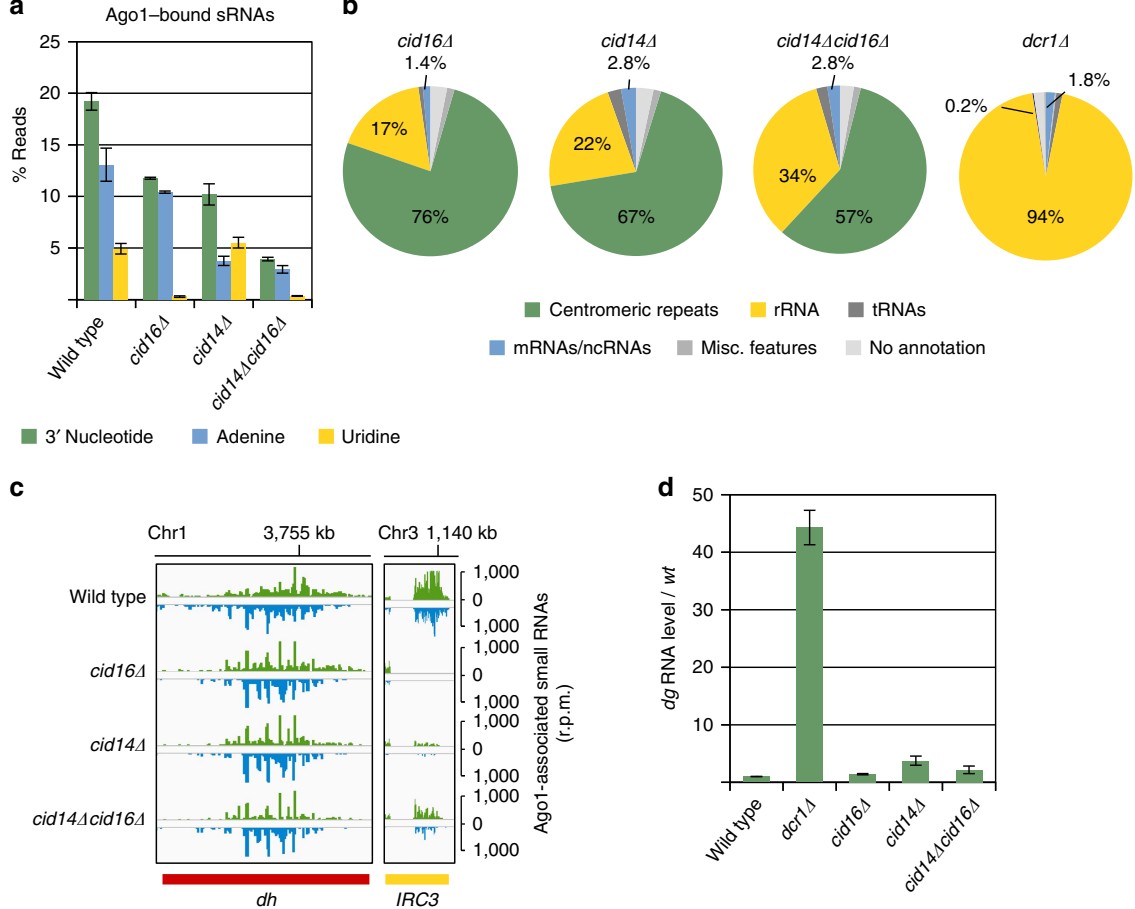

**Figure 2 | Cid14 adenylates and Cid16 uridylates the 3′ end of Argonaute-bound sRNAs.** (**a**) Quantification of Argonaute-bound sRNAs that have non-templated nucleotides at the 3′ end in indicated strains. Error bars indicate s.e.m. of two independent sRNA-sequencing experiments. Cid14 adenylates and Cid16 uridylates the 3′ end of sRNAs. (**b**) Argonaute-bound sRNAs were analysed by high-throughput sequencing from indicated cells and classified as indicated below the pie charts. Pie charts illustrate percentages for the individual sRNA classes relative to the total number of reads for each strain. (**c**) Argonaute-bound sRNA reads from indicated cells were plotted over centromeric region. The location of genes is indicated below the sRNA peaks. Reads from + and − strands are depicted in green and blue, respectively. Scale bars on the right denote sRNA reads numbers normalized per one million reads. (**d**) Quantification of centromeric *dg* transcripts in indicated strains by RT-qPCR. In *cid14Δ* and *cid14Δcid16Δ* cells *dg* RNA is accumulating. Error bars indicate s.e.m. of >six independent experiments. '/' indicates fold change.

rDNA and rRNA was more than 5-fold reduced in *cid14Δ* cells that over-express Rdp1 (Fig. 4e; Supplementary Fig. 4f). This indicates that RNAi silenced rRNA in these cells that would lead to observed slow growth phenotype. We observed that RNAi also targeted new mRNA targets in *cid14Δ* cells that over-express Rdp1 (Fig. 4f). Our data show that fission yeast cells use RNAi to restrict the RNAi machinery in response to improper processing of sRNAs. We found that *rdp1* mRNA is targeted by RNAi in several independent strains indicating that this provides rapid adaptation to uncontrolled RNAi. Our data show that fission yeast cells can use the RNAi machinery to reprogram their expression to adapt to stress conditions.

**Cid14 and Cid16 recruit Rrp6 to degrade Ago1-bound sRNAs.** The sRNA-sequencing data implicate that Cid14 and Cid16 add non-templated nucleotides to Argonaute-bound sRNAs (Fig. 2a). This prompted us to investigate if Argonaute interacts with Cid14 and Cid16. In a co-immunoprecipitation experiment, we observed that Argonaute and Cid14 interact *in vivo*, further supporting the observation that Cid14 adenylates Argonaute-bound sRNAs (Fig. 5a). Cid16 is expressed in low level, and we

could not detect Cid16 neither in the input nor in the immunoprecipitated fraction by the western blotting analysis.

Next, we purified the wild type and catalytically inactive Cid14 and Cid16 nucleotidyl-transferases (Supplementary Fig. 5a). In *in vitro* assays, Cid14 added 10–20 adenines and Cid16 added 1–3 uridines to free RNA (Supplementary Fig. 5b,c). Both Cid14 and Cid16 were specific in the addition of non-templated adenines and uridines, respectively (Supplementary Fig. 5d). Cid14 was also able to adenylate double stranded siRNA, while Cid16 showed strong reduction in activity on double stranded siRNA similarly to TUT4 in mammalian cells[10] (Supplementary Fig. 5e). Our sequencing and co-immunoprecipitation data indicate that Cid14 acts on Argonaute-bound sRNAs. Therefore, we loaded Argonaute with 22 nucleotides long single stranded sRNA[9]. Cid14 added either 1–3 or 15–20 non-templated adenines to the 3′ end of Argonaute-bound sRNAs, indicating that the Cid14 activity is modulated by Argonaute (Fig. 5b). We observed that 10–20% of 22 nucleotide long sRNAs dissociate from Argonaute in course of the assay[9], suggesting that 15–20 non-templated adenines are added to free sRNAs. This is consistent with the sequencing of Argonaute-bound sRNAs where we find 1–3 additional non-templated nucleotides at the 3′ end of sRNAs[8].

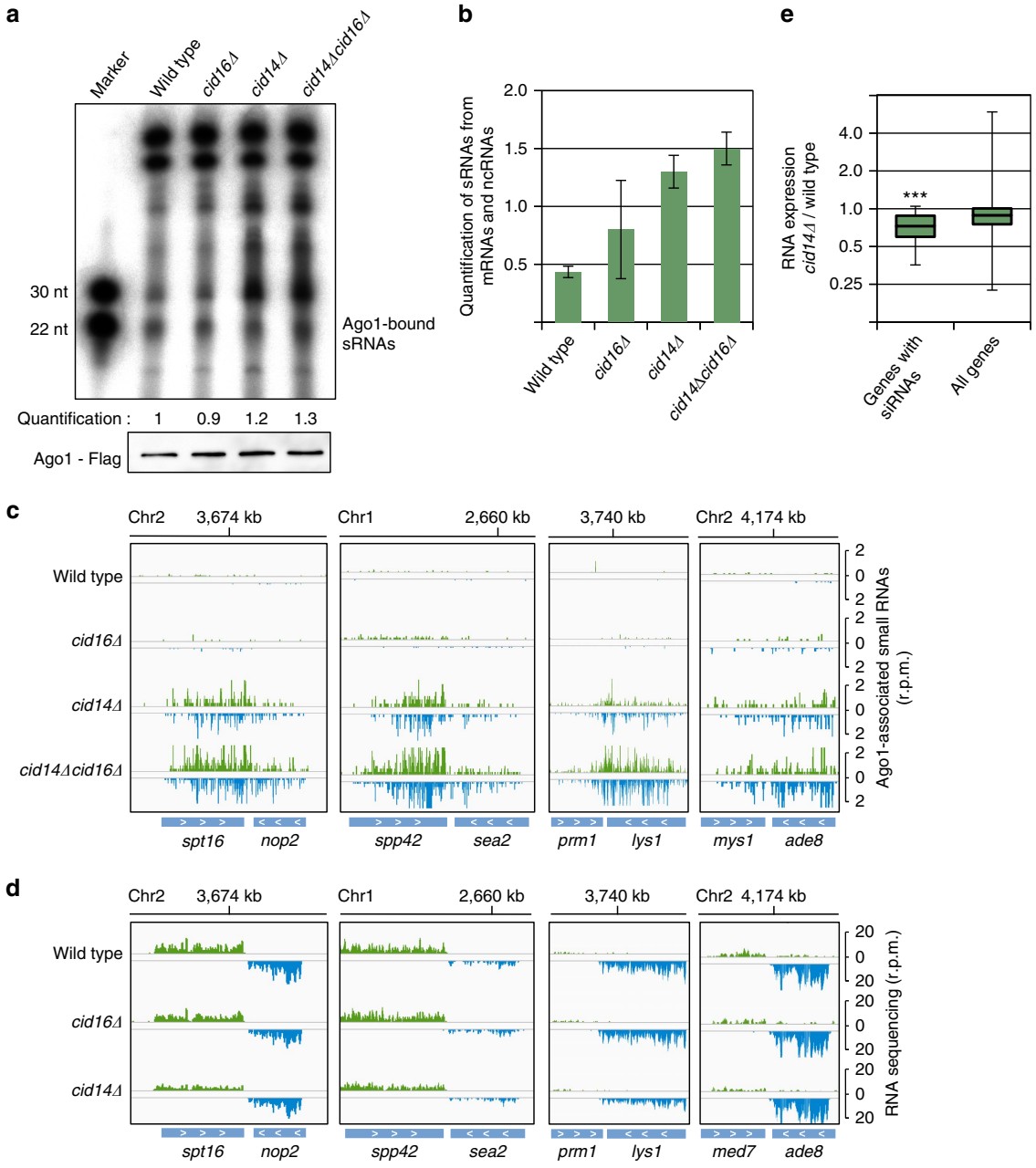

**Figure 3 | Accumulation of antisense priRNAs triggers uncontrolled RNAi in *cid14Δ* cells.** (**a**) Autoradiograph of denaturing polyacrylamide gel showing Argonaute-bound RNAs purified from wild type, *cid16Δ*, *cid14Δ* and *cid14cid16Δ* cells. Quantification is based on two independent biological replicates. Each band was normalized to higher unspecific bands of its lane and compared to wild type. Lower panels show western blotting detection of Flag-Argonaute protein in immunoprecipitation assays. (**b**) Quantification of euchromatic Argonaute-bound siRNAs and priRNAs mapping to mRNAs and ncRNAs in indicated strains. Euchromatic siRNAs and priRNAs accumulate in *cid16Δ*, *cid14Δ* and *cid14cid16Δ* cells. Error bars indicate s.e.m. of two independent sRNA-sequencing experiments. (**c**) Argonaute-bound sRNA reads from indicated strains were plotted over euchromatic genes. The location of genes is indicated below the sRNA peaks. Reads from + and − strands are depicted in green and blue, respectively. Scale bars on the right denote sRNA reads numbers normalized per one million reads. In *cid14Δ* and *cid14cid16Δ* cells, RNAi targets euchromatic genes. (**d**) RNA-sequencing reads in indicated cells are plotted over the several euchromatic genes. In *cid14Δ* cells genes that are targeted by RNAi are silenced. Scale bars on the right denote RNA reads numbers normalized per one million reads. (**e**) Box plot of differential expression of RNA in *cid14Δ* cells compared to wild-type cells for all genes and genes that generate siRNAs in *cid14Δ* cells. Genes that generate siRNAs show reduced level of mRNA in *cid14Δ* cells indicating silencing by RNAi. Two sided *t* test for two independent datasets with high variance was used to calculate the *P* value. \*\*\*$P < 1E-10$.

Similarly to Cid14, Cid16 added mostly 1–2 nucleotides to Argonaute-bound sRNAs in the *in vitro* assay (Fig. 5b).

Our previous work showed that Argonaute recruits Triman to process sRNAs to the final length[9]. We observed that adenines accumulated at the 3′ end of Argonaute-bound sRNAs in *tri1Δ* cells, but not in *rrp6Δ*, *dis3l2Δ* and *dis3-54* cells (Fig. 5c). This shows that Triman actively removes non-templated adenines from the 3′ end of sRNAs and it is consistent with Triman belonging to the PARN family of deadenylases[9]. Our data reveal that Cid14 and Triman act together to control the length and stability of sRNAs: Cid14 adenylates and Triman de-adenylates Argonaute-bound sRNAs. sRNA length is essential for Argonaute

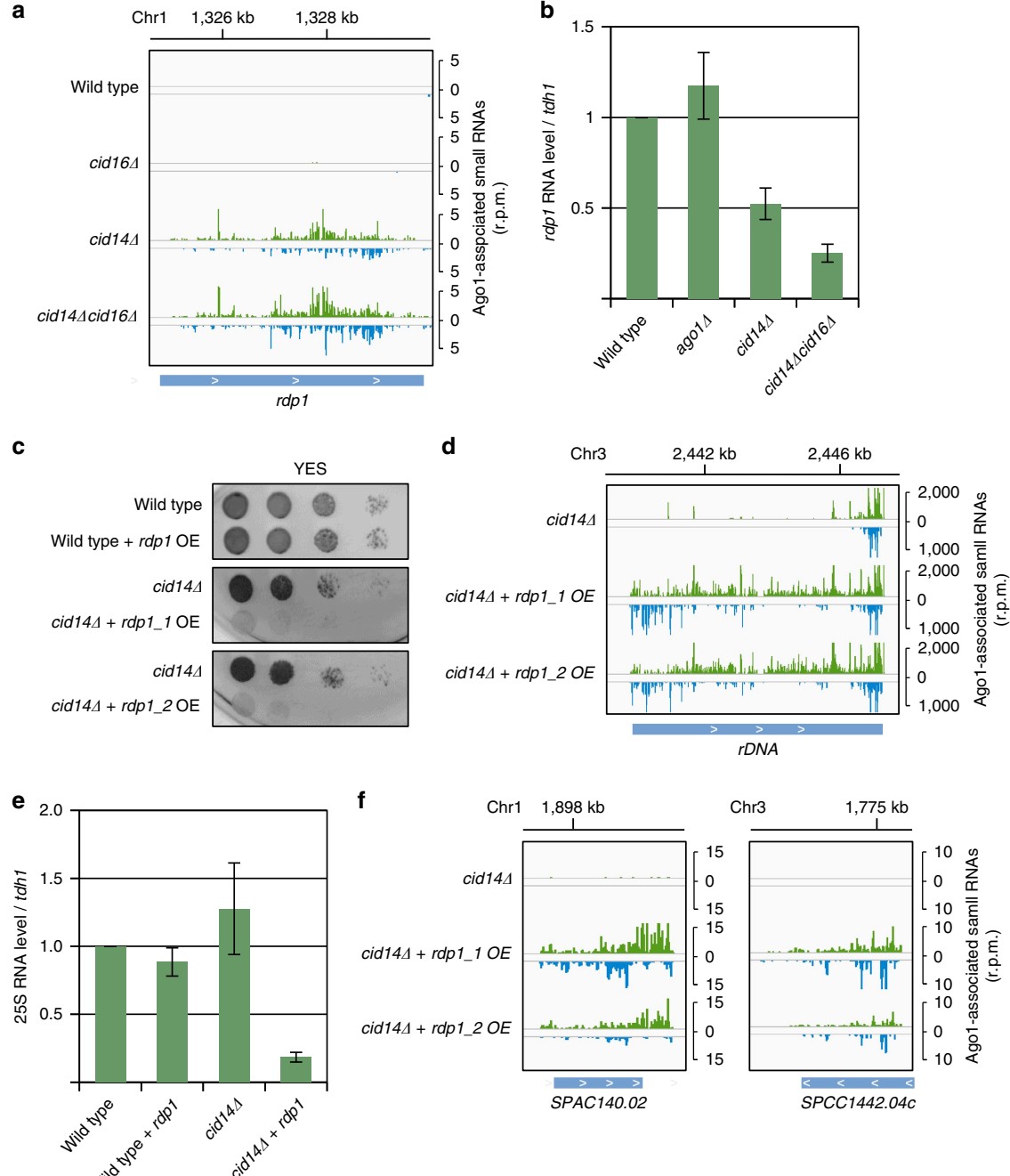

**Figure 4 | RNAi silences *rdp1* to reduce uncontrolled RNAi in *cid14Δ* cells. (a)** Argonaute-bound sRNA reads from indicated strains were plotted over *rdp1* gene. The location of genes is indicated below the sRNA peaks. Reads from + and − strands are depicted in green and blue, respectively. Scale bars on the right denote sRNA reads numbers normalized per one million reads. In *cid14Δ* and *cid14Δcid16Δ* cells, RNAi targets *rdp1*. **(b)** Quantification of *rdp1* transcripts in indicated strains by RT-qPCR. In *cid14Δ* and *cid14Δcid16Δ* cells *rdp1* mRNA is strongly reduced. Error bars indicate s.e.m. of three independent experiments. **(c)** Growth assay showing strong reduction in viability of *cid14Δ* cells that over-express *rdp1* gene. Cells were growing for 2 days before imaging. **(d)** Argonaute-bound sRNA reads from indicated strains were plotted over rDNA. Reads from + and − strands are depicted in green and blue, respectively. Scale bars on the right denote sRNA read numbers normalized per one million reads. In *cid14Δ* cell over-expressing Rdp1, much higher amount of siRNAs are generated at rDNA. **(e)** Quantification of 25S rRNA transcript in indicated strains by RT-qPCR. In *cid14Δ* cells that over-express *rdp1* gene, 25S rRNA is 5-fold reduced. Error bars indicate s.e.m. of three independent experiments. **(f)** Argonaute-bound sRNA reads from indicated strains were plotted over indicated genes. Reads from + and − strands are depicted in green and blue, respectively. Scale bars on the right denote sRNA reads numbers normalized per one million reads. In *cid14Δ* cells over-expressing Rdp1, RNAi targets new protein coding genes.

slicing activity[9] and addition or removal of non-templated nucleotides can change sRNA functionality. On the contrary, uridylation did not accumulate on Argonaute-bound sRNAs in *tri1Δ* cells, indicating that uridines are not removed by Triman

(Fig. 5c). Uridylated sRNAs also showed increased length suggesting that uridine(s) at the 3′ end protect sRNAs from trimming (Supplementary Fig. 5f). The longer sRNAs are not capable to slice their complementary targets and are functionally

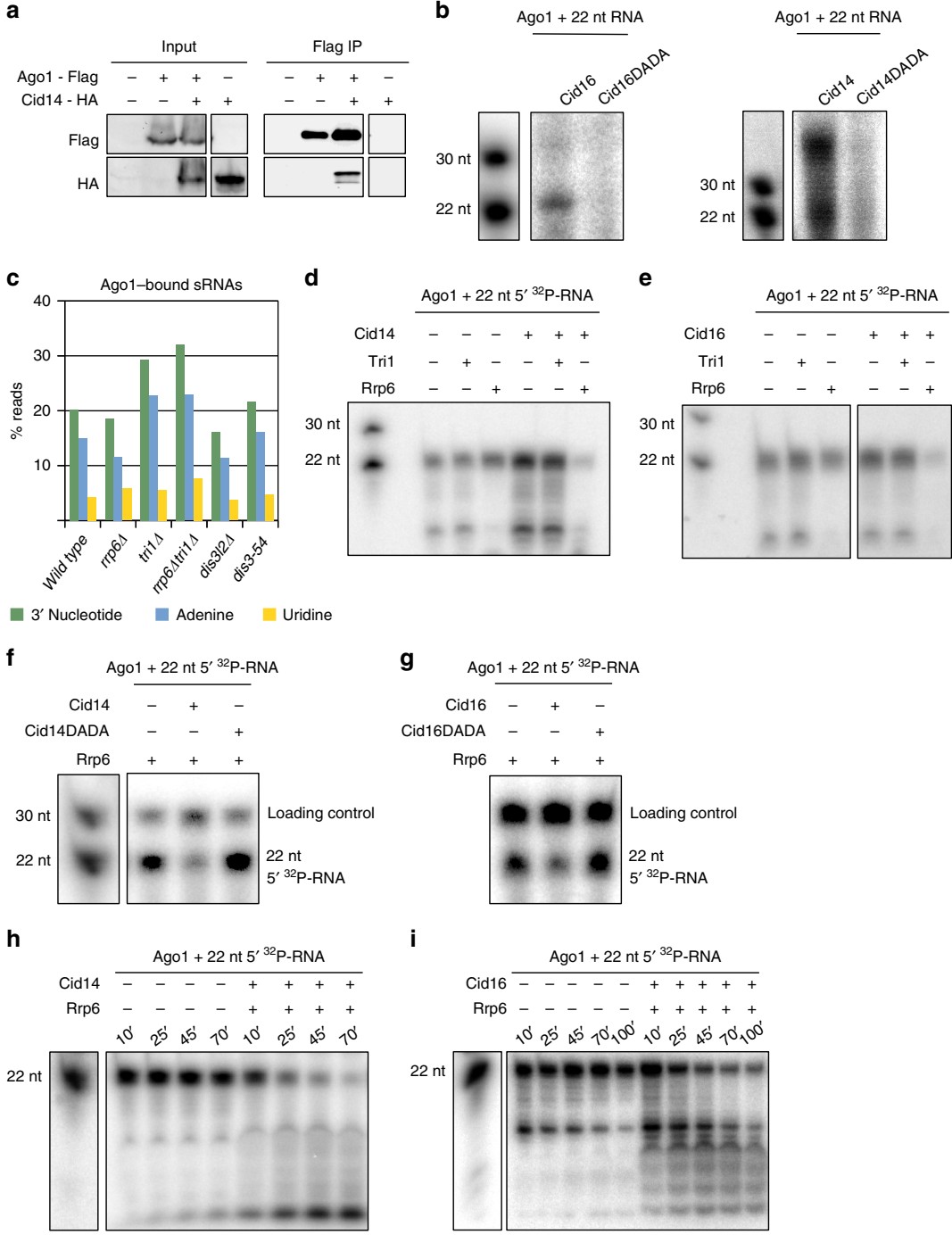

**Figure 5 | Cid14/Cid16 and Rrp6 degrade Argonaute-bound sRNAs. (a)** Western blotting analysis of co-immunoprecipitation assay showing that Argonaute interacts with Cid14 *in vivo*. (**b**) Autoradiograph of denaturing polyacrylamide gel showing Cid14 and Cid16 activity on Argonaute-bound sRNA. 22 nucleotide long sRNA was loaded onto empty Argonaute purified from *dcr1Δtri1Δ* cells[9]. Argonaute was incubated with Cid14/Cid16 and 32P α-ATP/α-UTP and sRNA was analysed on denaturing polyacrylamide gel. (**c**) Quantification of Argonaute-bound sRNAs that have non-templated nucleotides at the 3′ end in indicated strains. (**d,e**) Autoradiograph of denaturing polyacrylamide gel showing degradation of Argonaute-bound sRNA by Cid14, Cid16 and Rrp6. 5′ 32P-labelled 22 nucleotide long sRNA was loaded onto empty Argonaute purified from *dcr1Δtri1Δ* cells. Argonaute was incubated with Cid14, Cid16 and Triman or Rrp6. (**e**) Autoradiograph of denaturing polyacrylamide gel showing degradation of Argonaute-bound sRNA by Cid16 and Rrp6. 5′ 32P-labelled 22 nucleotide long sRNA was loaded onto empty Argonaute purified from *dcr1Δtri1Δ* cells. Argonaute was incubated with Cid16 and Triman or Rrp6. (**f**) Autoradiograph of denaturing polyacrylamide gel showing degradation of Argonaute-bound sRNA by Cid14 and Rrp6. 5′ 32P-labelled 22 nucleotide long sRNA was loaded onto empty Argonaute purified from *dcr1Δtri1Δ* cells. Argonaute was incubated with Rrp6, Cid14 and Cid14DADA. 30 nucleotide long DNA was used as a loading control. (**g**) Autoradiograph of denaturing polyacrylamide gel showing degradation of Argonaute-bound sRNA by Cid16 and Rrp6. 5′ 32P-labelled 22 nucleotide long sRNA was loaded onto empty Argonaute purified from *dcr1Δtri1Δ* cells. Argonaute was incubated with Rrp6, Cid16 and Cid16DADA. 30 nucleotide long DNA was used as a loading control. (**h,i**) Autoradiograph of denaturing polyacrylamide gel showing time course of Argonaute-bound sRNA degradation by Cid14/Cid16 and Rrp6. 32P-labelled 22 nucleotide long sRNA was loaded onto empty Argonaute purified from *dcr1Δtri1Δ* cells. Argonaute was incubated with Rrp6, Cid14 (**h**) or Cid16 (**i**). Time when reaction was stopped is indicated above the image.

inactivated[9]. Consistently, in mammalian cells depletion of TUT4/7 uridyl-transferases does not affect miRNA abundance but reduces their activity[10,33].

Next, we wanted to test if adenylation or uridylation of Argonaute-bound sRNAs promotes their degradation. Cid14 is a member of the TRAMP complex, which recruits the nuclear exosome to targeted RNAs and promotes their degradation[24,34–36]. We loaded 22 nucleotide long 5′ $^{32}$P-labelled sRNA onto Argonaute and added the Cid14/Cid16 nucleotidyl-transferases and Triman or the 3′–5′ exonuclease Rrp6 to the reaction (Fig. 5d,e). Neither Triman or Rrp6 were able to remove the 22 nucleotide long sRNA from Argonaute[9], although Rrp6 efficiently degraded short contaminating RNA that is not bound by Argonaute (Fig. 5d,e). In presence of Cid14 or Cid16, however, Rrp6 degraded Argonaute-bound sRNAs (Fig. 5d,e). Nucleotidyl-transferase activity of both Cid14 and Cid16 was required for degradation of Argonaute-bound sRNAs (Fig. 5f,g). When we used catalytically dead mutants Cid14DADA and Cid16DADA, Argonaute-bound sRNAs were not degraded in in vitro assays. We also performed the time course experiments that show appearance of intermediate degradation products (Fig. 5h,i).

We had previously shown that Triman trims Argonaute-bound sRNAs to 22 nucleotides[9], however, we do not observe removal and degradation of Argonaute-bound sRNAs by Triman (Fig. 5d,e). We show that in vitro adenylation and uridylation of Argonaute-bound sRNAs by Cid14 and Cid16 recruits the 3′–5′ exonuclease Rrp6 to remove and degrade sRNAs from Argonaute. This suggests that adenylation by Cid14 can recruit either Triman or Rrp6 in vivo. While Rrp6 will degrade Argonaute-bound sRNAs, Triman will trim them to 22 nucleotides and make them functional again.

**Ago1-bound sRNAs are more stable in cid14Δ cells.** To determine degradation rate of Argonaute-bound and total sRNAs in wild type and cid14Δ cells, we generated a construct expressing ura4 hairpin[37] under the control of a repressible nmt1 promoter. In absence of thiamine, cells will express ura4 hairpin, which will be processed by Dicer to generate ura4 siRNAs (Supplementary Fig. 6a). Addition of thiamine represses nmt1 promoter and reduces ura4 mRNA levels in both wild type and cid14Δ cells (Supplementary Fig. 6b). This allowed us to follow the degradation of Argonaute-bound and total ura4 sRNAs in wild type and cid14Δ cells (Fig. 6a). ura4 sRNAs were not detectable in dcr1Δ or dcr1Δcid14Δ cells showing that they are genuine siRNAs (Supplementary Fig. 6c).

In wild-type cells, we observed a rapid degradation of ura4 siRNAs in total sRNA fraction, which includes Argonaute-bound and free sRNAs. Majority of ura4 siRNAs were degraded in < 5 h (Fig. 6a). In cid14Δ cells, after initial rapid degradation, we observed a reduction in degradation of ura4 siRNAs when purified from a total sRNA fraction (Fig. 6a,b). This suggests that free ura4 siRNAs are rapidly degraded independently of Cid14, while Argonaute-bound siRNAs might be stabilized in cid14Δ cells.

We looked at the degradation of Argonaute-bound sRNAs in wild type and cid14Δ cells. We observed that in wild-type cells, Argonaute-bound sRNAs are degraded slower than total sRNAs showing that binding to Argonaute increases the half-life of sRNAs (Supplementary Fig. 6e). In cid14Δ cells, we observed only a small reduction in Argonaute-bound sRNAs 5 h after expression of ura4 hairpin was repressed (Fig. 6c,d; Supplementary Fig. 6d). This shows that Argonaute-bound sRNAs have longer half-life in cid14Δ cells when compared to the wild-type cells. Degradation of Argonaute-bound ura4 siRNAs was reduced much more than

degradation of ura4 siRNA from total sRNA fraction in cid14Δ cells (Supplementary Fig. 6f). This shows that Cid14 adenyl-transferase is mainly required for degradation of Argonaute-bound sRNAs, consistent with our finding that only Argonaute-bound sRNAs are tailed at the 3′ end.

We show that degradation of Argonaute-bound sRNAs is necessary to reduce accumulation of 'noise' sRNAs bound to Argonaute and to protect the genome from uncontrolled RNAi (Fig. 7).

**Discussion**

Eukaryotes have evolved various surveillance mechanism to monitor the quality of RNAs. Addition of non-templated nucleotides to the 3′ end of sRNAs is conserved in most organisms and promotes their degradation commonly before loading onto Argonaute. In Drosophila, however, Wispy was shown to adenylate Argonaute-bound maternally deposited mature miRNA which are then eliminated by a yet unknown nuclease(s)[19]. This suggested that mature sRNAs could be actively removed from Argonaute. We show here, that the non-canonical nucleotidyl-transferase Cid14 adenylates and Cid16 uridylates Argonaute-bound sRNAs. Both Cid14 and Cid16 are able to recruit Rrp6 in vitro to degrade Argonaute-bound sRNAs (Fig. 7a).

Consistent with our data on Cid16, TUT4/7 uridyl-transferases are specific to single stranded sRNAs and were also suggested to uridylate mature miRNA[10,33]. In agreement with our results, uridylation by TUT4/7 was also shown to facilitate pre-miRNA degradation by the exosome[18] (Fig. 5e). Our data show that fission yeast Cid16 is a uridyl-transferase that is a functional homologue of the TUT4/7 nucleotidyl-transferases in mammalian cells.

Cid16 was suggested to be localized to cytoplasm[21] indicating that sRNAs might be uridylated in the cytoplasm and imported to nucleus where they can be degraded by Rrp6. This might explain why we observe stronger uridylation of sRNAs originating from mRNAs (Fig. 1f) which are more abundant in the cytoplasm. This suggests that mRNA degradation products are loaded on Argonaute during its import to the nucleus where they are removed by the exosome. In contrast, Cid14 is localized in the nucleus[21], consistent with higher adenylation of sRNAs that are generated from ncRNAs that are predominately degraded in the nucleus. This suggests that uridylation and adenylation are spatially separated.

Our data show that all classes of sRNAs are modified by Cid14 and Cid16, and actively removed from Argonaute. The turnover will, however, mostly affect the least abundant sRNAs that are produced at slowest rate. This will put them effectively below the threshold required to generate secondary siRNAs and initiate RNAi. Consistent with this, we do not observe serious defects in generation of siRNAs at centromeric repeats. At centromeric repeats new siRNAs are constantly and efficiently generated and this overcomes their degradation. In cid14Δ, cid14Δcid16Δ and rrp6Δ[9] cells, we observe accumulation of euchromatic priRNAs that are normally removed by Cid14 and Rrp6. These priRNAs recruit RNAi to ectopic targets in these cells, including essential genes. Consistent with this, we had previously observed uncontrolled RNAi also in rrp6Δ cells[9,38]. In deletion of the nuclear exosome Rrp6 many non-coding RNAs accumulate[39]. It is likely that in rrp6Δ cells uncontrolled RNAi is induced by a cumulative effect of accumulation of antisense transcripts and failed turnover of Argonaute-bound sRNAs[9,38]. Cid14 and the TRAMP complex have, however, little effect on degradation of non-coding RNAs in fission yeast[28]. Also, our in vivo data show that Argonaute-bound sRNAs have increased half-life in cid14Δ cells. This shows that uncontrolled RNAi appearing in cid14Δ

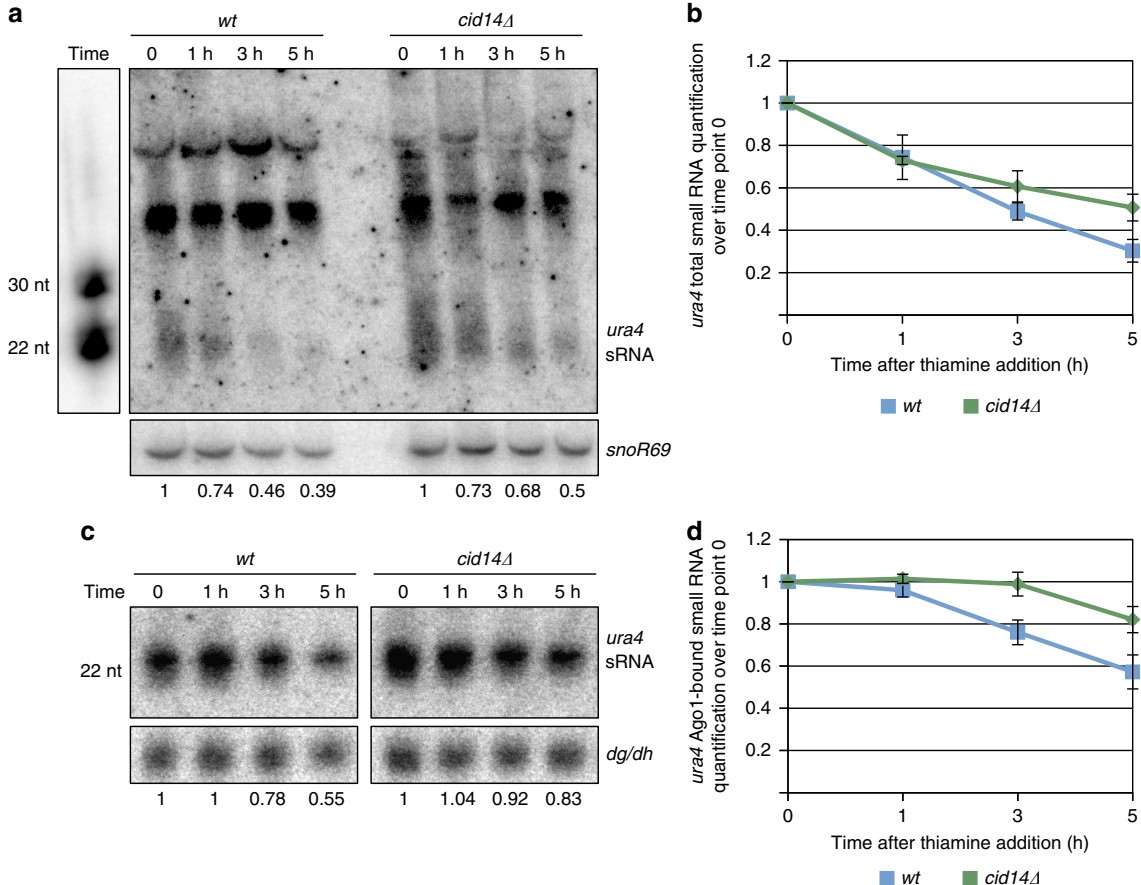

**Figure 6 | Argonaute-bound sRNAs are more stable in *cid14Δ* cells.** (**a**) Northern blotting showing *ura4* sRNAs isolated from total sRNA fraction from the indicated strains. *ura4* sRNAs were normalized to *snoR69* shown in the lower panel. Quantification is relative to time point 0 when thiamine was added and is shown below the image. (**b**) Quantification of *ura4* sRNAs from total sRNA fraction in wild type and *cid14Δ* cells. Quantification is relative to time point 0 when thiamine was added. *ura4* sRNAs were normalized to *snoR69*. Error bars indicate s.e.m. of three independent experiments. (**c**) Northern blotting showing Argonaute-bound *ura4* sRNAs isolated from wild type and *cid14Δ* cells. In *cid14Δ* cells, *ura4* sRNAs have longer half-life than in the wild-type cells. Argonaute-bound *ura4* sRNAs were normalized to centromeric sRNAs shown in the lower panel. Quantification is relative to time point 0 when thiamine was added and is shown below the image. (**d**) Quantification of Argonaute-bound *ura4* sRNAs in wild type and *cid14Δ* cells. *ura4* sRNAs were normalized to centromeric siRNAs. Quantification is relative to time point 0 when thiamine was added. Error bars indicate s.e.m. of three independent experiments.

and *cid14Δcid16Δ* cells is the result of a defect in sRNA turnover on Argonaute.

We observed siRNA generation at many euchromatic genes, however, we did not detect H3K9 methylation at these genes. Our data show that siRNA generation and H3K9 methylation can be uncoupled in fission yeast. Deficiency in degradation of Argonaute-bound sRNAs recruits RNAi to ectopic targets and generates siRNAs that silence mRNA targets without establishing H3K9 methylation. This suggests that siRNAs at ectopic places are not generated on chromatin but target RNAs post-transcriptionally. Even when we over-expressed Rdp1, we did not observe H3K9 methylation at euchromatic genes, although higher amounts of siRNAs were generated. These data show that chromatin remains refractory to H3K9 methylation and hetero-chromatin formation in *cid14Δ* and *cid14Δcid16Δ* cells, and indicate that changes in chromatin organization are required for heterochromatin formation.

Uncontrolled RNAi that targets random genes required for normal cellular function is clearly not advantageous for the growth of cells. We found that in *cid14Δ* and *cid14Δcid16Δ* cells RNAi targets the *rdp1* gene. Interestingly, we observed the same *rdp1* silencing in *cid14Δ*, *cid14Δcid16Δ* and *rrp6Δ* cells. Rdp1 is an RNA-dependent RNA polymerase that is required for dsRNA synthesis and siRNA generation, and it is essential for

RNAi[8,30,32]. We observed a strong reduction in *rdp1* mRNA abundance, which reduces the efficiency of the RNAi machinery and protects the genome from even more deleterious RNAi (Fig. 7b). Consistent with this, over-expression of Rdp1 in *cid14Δ* cells strongly reduces their viability, indicating that the re-programing of *rdp1* expression is essential for viability. It is likely that *rdp1* was targeted randomly by RNAi, and cell, that silenced *rdp1*, cell was selected as fittest in this evolutionary experiment. It seems that silencing of *rdp1* can provide a balance between functional centromeric RNAi and restricted ectopic RNAi. A similar protection mechanism was recently reported for *epe1Δmst2Δ* (*epe1*, putative histone demethylase; *mst2*, histone acetyltransferase) fission yeast cells[40]. These cells accumulated H3K9 methylation at the locus of the H3K9 methyltransferase Clr4 to prevent uncontrollable spreading of heterochromatin. Our data show that yeast cells can also use RNAi to reprogram their genome expression to adapt to external or internal stresses[41]. Compared to genetic mutations, RNAi-based adaptations provide faster and reversible responses, allowing an easy reversion to normal transcription profiles when external stimuli disappear. In cancer cells, epigenetic variations might enable tumour cells to adapt to stress conditions and to survive therapies[42,43].

In this study, we show that Cid14 and Cid16 add non-templated nucleotides to Argonaute-bound sRNAs and promote

**a**

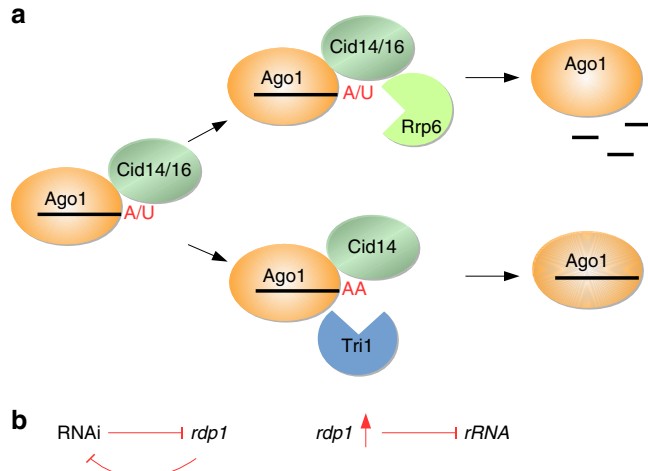

**b**

RNAi ⊣ *rdp1*     *rdp1* ↑ ⊣ *rRNA*

**Figure 7 | Cid14/Cid16 and Rrp6 degrade Argonaute-bound sRNAs to protect the genome from uncontrolled RNAi.** (**a**) Cid14 adenylates and Cid16 uridylates Argonaute-bound sRNAs in fission yeast. Both Cid14 and Cid16 recruit the Rrp6 nuclease to actively degrade sRNAs from Argonaute. Adenylation by Cid14 can recruit either Triman or Rrp6. Rrp6 will degrade Argonaute-bound sRNA. To the contrary, Triman will trim Argonaute-bound sRNAs to 22 nucleotides and make them functional again. (**b**) Degradation of Argonaute-bound sRNAs is necessary to reduce accumulation of 'noise' sRNAs in Argonaute and to protect the genome from uncontrolled RNAi. In *cid14Δ* and *cid14Δcid16Δ* cells RNAi targets the *rdp1* gene and suppresses itself. Over-expression of Rdp1 in *cid14Δ* cells silences rRNA and many euchromatic genes and strongly impairs the growth. This indicates that the re-programing of *rdp1* expression is essential for viability of *cid14Δ* and *cid14Δcid16Δ* cells.

their degradation by the exosome *in vitro*. Our data suggest that tailing and degradation of Argonaute-bound sRNAs protect the genome from uncontrolled RNAi. miRNAs are essential in regulating gene expression in multicellular organisms[44]. Degradation of Argonaute-bound miRNAs may be necessary to exchange miRNAs from Argonaute during cell differentiation and development. A defect in turnover of Argonaute-bound sRNAs might also lead to uncontrolled silencing and tumorigenesis in mammalian cells. It will be interesting in the future to investigate tailing and degradation of Argonaute-bound sRNAs in various mammalian cell types and during cell differentiation, development and tumorigenesis.

## Methods

**Strain construction and plasmid generation.** All *S. pombe* strains used in this study are listed in Supplementary Table 1. The strains were constructed by electroporation (Biorad MicroPulser programme ShS) with plasmid or a PCR-based gene targeting product leading to deletion or epitope-tagging of specific genes. Positive transformants were selected on YES plates containing 100–200 mg ml$^{-1}$ antibiotics and were confirmed by PCR and sequencing. Strains containing plasmids (Supplementary Table 2) were grown on Complete Edinburgh Minimal Medium-Leu. For endogenous C-terminal tagging, plasmid p85 harbouring 3xHA tag (Supplementary Table 3) was used together with primers 523 and 524 (Supplementary Table 4) for Cid14 and Cid16, respectively. *cid14* was cloned into pRSF-Duet with a GST tag on the N terminal and a six histidine tag on the C terminal was inserted via inverse PCR. *cid16* was cloned into pREP1 with a Flag tag on the N terminal and a six histidine tag on the C terminal was inserted via inverse PCR. The *ura4* hairpin[37] was excised from the original plasmid pnatMX ART sh ura4-5 and ligated into pREP1 after a double digestion with XmaI and NdeI. See Supplementary Tables 3 and 4.

**Ago1-bound siRNA sequencing.** Endogenous 3xFLAG-tagged Ago1 was purified from different mutants by protein affinity purification. The pellet of a 2.5 l log-phase culture was resuspended 1:1 in lysis buffer (50 mM HEPES pH 7.5, 1.5 M NaOAc, 5 mM MgCl$_2$, 2 mM EDTA pH 8, 2 mM EGTA pH 8, 0.1% Nonidet P-40, 20% Glycerol) containing 1 mM PMSF, 0.8 mM DTT and complete EDTA-free

Protease Inhibitor Cocktail (Roche). Cells were lysed with 0.25–0.5 mm glass beads (Roth) using the BioSpec FastPrep-24 bead beater (MP-Biomedicals) (4 cycles of 30 s at 6.5 m s$^{-1}$ then 5 min on ice). The lysate was spun at 13,000g for 15 min to remove cell debris. The supernatant was incubated with 30 µl Flag-M2 affinity gel (Sigma, A2220) for 1.5 h at 4 °C. The resin was washed 5 times with lysis buffer. Ago1 was eluted with 1% SDS, 300 mM NaOAc. The protein-bound RNA was recovered by phenol–chloroform–isoamylalcohol (25:24:1, Roth) extraction and ethanol precipitation. sRNAs with the length 20–30 nt were excised from an 18% acrylamide urea gel. 2 pmol of a preadenylated 3′ adaptor oligonucleotide (miRNA Cloning Linker-1 from IDT, 5′-App CTGTAGGCACCATCAAT/ddC/-3′) were ligated in a 10 µl reaction with 5 U T4 RNA ligase (TaKaRa), ligation buffer without ATP and 5 U RNasin (Promega) at 20 °C for 2 h. The 3′ ligated products were purified on an 18% acrylamide urea gel with subsequent phenol-chloroform purification and ethanol purification. The 5′ adaptor ligation was performed in a 10 µl reaction with 2 pmol 5′ adaptor oligonucleotide (5′-GUUCAGAGUUCUA-CAGUCCGACGAUC-3′), 5 U RNasin (Promega), 0.06 µg BSA, 5 U T4 RNA ligase (Thermo Scientific) and 1× ligation buffer with ATP (Thermo Scientific) for 2 h at 20 °C. The ligated products were gel purified and reverse transcribed with 10 pmol primer (RT primer: 5′- GTGACTGGAGTTCAGACGTGTGCTCTTCCGATCG ATTGATGGTGCCTACAG-3′) and the SuperScript III First Strand Synthesis System (Thermo Scientific). The cDNA was PCR-amplified with Q5 High-Fidelity 2× Master Mix (NEB) for 14–20 cycles using the Illumina P5 5′ primer (5′-AAT GATACGGCGACCACCGAGATCTACACTCTTTCCCTACACGACG-3′) and the Illumina P7 3′ primer with inserted barcode (5′-CAAGCAGAAGACGGCATA CGAGATXXXXXXGTGACTGGAGTTCAGACGTG-3′). Single end sequencing was performed on an Illumina GAIIX sequencer at the LAFUGA core facility of the Gene Center, Munich. The Galaxy platform was used to demultiplex the obtained reads[45]. Total size selected sRNAs (GEO: GSE19734) and Ago1-bound sRNAs from wild type, *cid14Δ*, *cid16Δ* and *cid14Δcid16Δ* were sequenced twice.

**Total RNA isolation.** Total RNA was isolated from mid-log phase yeast culture with the TRI Reagent Solution (Ambion) according to the manufacturer's instructions. DNAse I (Thermo Scientific) treatment was performed for 1–2 h at 37 °C. DNAse was removed by a second phenol–chloroform–isoamylalcohol extraction and total RNA was precipitated with ethanol. Total sRNA fraction was enriched via column purification Rneasy kit (Qiagen) according to the manufacturer's instructions.

**Reverse transcription and quantitative real-time PCR.** Overall, 250 ng of total RNA was reverse transcribed with SuperScript III First Strand Synthesis System (Thermo Scientific) and 0.2 pmol specific primers (Supplementary Table 4). Real-time PCR was performed with 3.5 ng of cDNA, DyNamo Flash SYBR Green qPCR Master Mix (Thermo Scientific), and specific primers in the Toptical thermocycler (Biometra), according to the manufacturer's instructions. qRT-PCR was perfomed in triplicate for each cDNA sample and primer. A non reverse-transcribed sample was used as control for DNA contamination. The experiments were performed at least in three biological replicates.

**Total RNA sequencing.** rRNA of 1 µg total RNA was degraded with Terminator nuclease (Epicentre) in buffer A at 30 °C for 2 h. The RNA library was obtained using the NEBNext Ultra Directional RNA Library Prep Kit for Illumina (NEB). Single end sequencing was performed on an Illumina GAIIX sequencer at the LAFUGA core facility of the Gene Center, Munich. The Galaxy platform was used to demultiplex the obtained reads. The sequencing was performed once and some datasets were confirmed by RT-qPCR in at least three biological replicates.

**Growth assay.** Tenfold serial dilutions of cultures with OD$_{600}$ between 0.4 and 0.7 were made so that the highest density spot contained 10$^5$ cells. Cells were spotted on non-selective YES medium. The plates were incubated at 32 °C for 2–3 days and imaged.

**Chromatin immunoprecipitation.** Overall, 50 ml mid-log phase yeast cultures with were cross-linked with 1% formaldehyde (Roth) for 15 min at room temperature. The reaction was quenched with 125 mM glycine for 5 min. The frozen pellet was resuspended in 500 µl lysis buffer (250 mM KCl, 1× Triton-X, 0.1% SDS, 0.1% Na-Desoxycholate, 50 mM HEPES pH 7.5, 2 mM EDTA, 2 mM EGTA, 5 mM MgCl2, 0.1% Nonidet P-40, 20% Glycerol) with 1 mM PMSF and Complete EDTA-free Protease Inhibitor Cocktail (Roche). Lysis was performed with glass beads (Roth) and the BioSpec FastPrep-24 bead beater (MP-Biomedicals), 8 cycles at 6.5 m s$^{-1}$ for 30 s and 3 min on ice. DNA was sheared by sonication (Bioruptor, Diagenode) 35 times for 30 s with a 30 s break. Cell debris was removed by centrifugation at 13,000g for 15 min. The crude lysate was normalized based on the RNA and Protein concentration (Nanodrop, Thermo Scientific) and incubated with 1.2 µg immobilized (Dynabeads Protein A, Thermo Scientific) antibody against dimethylated H3K9 (H3K9me2, Abcam AB1220) for 2 h or overnight at 4 °C. The resin with immunoprecipitates was washed five times with each 1 ml of lysis buffer and eluted with 150 µl of elution buffer (50 mM Tris HCl pH 8, 10 mM EDTA, 1% SDS) at 65 °C for 15 min. RNase A (Thermo Scientific) treatment was

performed for 20 min at 65 °C and subsequent Proteinase K (Roche) treatment was performed for at least 5 h or ON at 65 °C. DNA was recovered by phenol–chloroform–isoamylalcohol (25:24:1, Roth) extraction with subsequent ethanol precipitation. DNA levels were quantified by qRT-PCR and normalized to *tdh1* background levels. Oligonucleotides used for quantification are listed in Supplementary Table 4. For sequencing, a ChIPseq library was made using the NEBNext Ultra II DNA Library Prep Kit for Illumina kit (NEB).

**Analysis of sequencing data.** Demultiplexed illumina reads were mapped to the S. *pombe* genome, allowing 2 nucleotides mismatch to the genome using Novoalign (htttp://www.novocraft.com). sRNAs reads mapping to multiple locations were randomly assigned. The datasets were normalized to the number of reads per million sequences for sRNAseq and ChIPseq or reads mapping to coding sequences for total RNAseq. In addition, the datasets were normalized to total amounts of reads (for sRNAs) that were associated with Ago1 in different mutant strains as determined by Ago1 pulldowns and quantification of bound sRNAs. We used the genome sequence and annotation that were available from the S. *pombe* Genome Project[46]. The data are displayed using the Integrative Genomics Viewer (IGV)[47]. Sequenced strains are listed in Supplementary Table 5.

**Statistical analysis.** Two sided *t* test for two independent datasets with high variance was used to calculate the *P* value.

**siRNA purification and detection.** siRNAs associated with Argonaute were purified as described in Marasovic et al.[9] Briefly, siRNAs were recovered from FLAG-Ago1 by phenol–chloroform extraction and ethanol precipitation, labelled with $[\gamma^{32}-P]$-ATP and run on 18% denaturing polyacrylamide gels. The gel was wrapped in cling film and exposed to a storage phosphor screen (BAS MS 2025—Fujifilm Corporation) overnight up to 2 days at $-80$ °C. The screen was scanned with a Typhoon FLA 9500 (GE Healthcare). The experiment was performed in two biological replicates.

**Protein expression and purification.** Flag-Rrp6 (kindly provided by François Bachand) and GST-Cid14-6xHis were expressed in *E. coli* with 0.2 mM IPTG at 18 °C overnight. Pelleted cells were resuspended in lysis buffer (50 mM $NaH_2PO_4/$ $Na_2HPO_4$ pH = 8, 1 M NaCl, 20 mM imidazole, 3 mM b-mercaptoethanol and 0.5 mM PMSF) and lysed with the French Press and the clear lysate was incubated with Ni Sepharose 6 Fast Flow (GE Healthcare) for 30 min at 4 °C. The resin was washed three times with 50 ml of lysis buffer, once with 4 resin volume of 40 mM imidazole lysis buffer and the protein was eluted with 6 resin volume of elution buffer (50 mM $NaH_2PO_4/Na_2HPO_4$ pH 8, 500 mM NaCl, 300 mM imidazole, 3 mM b-mercaptoethanol and 1 mM PMSF). The elution fraction was dialysed in 50 mM Tris pH 7.5, 150 mM NaCl, 0.1 mM DTT, 0.1 mM EDTA and incubated with Glutathione Sepharose 4 Fast Flow (GE Healthcare) for 30 min at 4 °C. The resin was washed twice with dialysis buffer and the protein was eluted with 6 × column volume of elution buffer (50 mM Tris pH 8, 500 mM NaCl, 10 mM reduced glutathione). The elution fraction was dialysed in 50 mM Tris pH 7.5, 200 mM NaCl, 1 mM DTT, 0.1 mM EDTA, 10% glycerol). Flag-Cid16-6xHis was expressed in *S. pombe*. Pelleted cells were resuspended in lysis buffer (50 mM HEPES pH 7.5, 1.5 M NaOAc, 5 mM $MgCl_2$, 2 mM EDTA pH 8, 2 mM EGTA pH 8, 0.1% Nonidet P-40, 20% Glycerol) containing 1 mM PMSF, 0.8 mM DTT and Complete EDTA-free Protease Inhibitor Cocktail (Roche). Frozen cells were lysed with freezer mill and the protein was purified with Ni-NTA resin as described above. The dialyzed elution fraction was incubated with Flag-M2 agarose beads for 2–3 h at 4 °C. The beads were washed with 20 column volumes of dialysis buffer. Protein was eluted with 2 × column volum of elution buffer with 0.2 mg ml$^{-1}$ 3xFlag peptide. Rrp6 was purified with Flag-M2 agarose beads as described above.

**Co-immunoprecipitation.** Flag immunoprecipitation experiments were performed as described above but in low salt conditions (50 mM HEPES pH 7.5, 100 mM NaOAc, 5 mM $MgCl_2$, 2 mM EDTA pH 8, 2 mM EGTA pH 8, 0.1% Nonidet P-40, 20% Glycerol). All samples and corresponding inputs were analysed by immuno-blot as described in Marasovic et al.[9] Protein were separated on an 8% polyacrylamide SDS-page and transferred on a PVDF membrane (Roth Immobilon-P) for 1 h at 15 voltage using a Trans-Blot SD Semi-Dry Transfer Cell (BioRad). The membrane was blocked with 5% milk (w/v) in 1 × TBS-T buffer (50 mM Tris-Cl pH 7.5, 150 mM NaCl, 0.05% Tween-20) for 1 h at room temperature, incubated with the Anti-HA antibody (Santa Cruz Biotechnology, sc-7392, 1:200 in 1 × TBS-T) for 1 h and washed three times with 1 × TBS-T for 10 min at room temperature. The membrane was then incubated with the secondary anti-mouse antibody coupled to peroxidase (BioRad, #1721011, 1:3,000 in 1 × TBS-T) for 1 h at room temperature and washed three times with 1 × TBS-T for 10 min.

The membrane was developed using the Super Signal West Pico Chemiluminescence Substrate. The membrane was incubated with 20 ml of Restore Western Blot Stripping Buffer (Thermo Fisher Scientific) for 20 min at room temperature, blocked with 5% milk (w/v) in 1 × TBS-T for 1 hour at room temperature and incubated with the peroxidase conjugated Anti-FLAG antibody (Sigma-Aldrich, #8592, 1:1,000 in 1 × TBS-T) for 1 h at room temperature.

**Degradation of Argonaute-associated sRNAs *in vitro*.** Argonaute was purified from dcr1Δtri1Δ cells as described above, with the exception that Ago1 remained associated with the FLAG resin. Ago1 associated with the resin was incubated with 500 fmol of $^{32}P$ radiolabeled small RNAs for 2 hr. Resin was washed with buffer containing 25 mM HEPES pH 7.5, 2 mM $MgCl_2$, 2 mM DTT, 0.02% NP-40, and 100 mM NaOAc to remove unbound small RNAs.

**Northern blotting analysis.** Overall, 2–5 µg of total sRNAs were run on 18% denaturing polyacrylamide gel and transferred to a positively charged nylon membrane (GE Healthcare Amersham Hybond N +) on a Trans-Blot SD Semi-Dry Transfer Cell (Biorad). The RNA was ultraviolet-cross-linked to the membrane with Spectrolinker XL-1500 (Spectroline, 'optimal crosslink'). Prehybridization was performed with Church Buffer (0.5 M $NaH_2PO_4/Na_2HPO_4$ pH 7.2, 1 mM EDTA, 7% SDS) at 37 °C overnight. 10 pmol of DNA probes (Supplementary Table 4) were labelled with T4 PNK (NEB) and 10 pmol $[\gamma-^{32}P]$-ATP (Hartmann Analytic) at 37 °C for 60 min. The labelled probes were purified with an Illustra MicroSpin G-25 column (GE Healthcare), mixed with 5 ml Church Buffer, and incubated with the membrane for 5 h at 37 °C. The membrane was rinsed once with 2x SSC buffer (0.3 M NaCl, 0.03 M sodium citrate, pH 7) and then washed three times with 2x SSC buffer for 15 min at 37 °C. The membrane was wrapped in cling film and exposed to a storage phosphor screen (BAS MS 2025—Fujifilm Corporation) overnight up to 2 days at $-80$ °C. The screen was scanned with a Typhoon FLA 9500 (GE Healthcare). For a second hybridization of the same membrane, the membrane was stripped in boiling 0.1% SDS for 5 min and subsequently prehybridized.

**sRNAs tailing assay.** Overall, 500 fmol to 1 pmol of double-strand or single strand 22 nucleotides long RNAs were incubated with 80 ng of Cid14 and Cid16 in buffer containing 1 mM Hepes pH 7.5, 0.5 mM $MgCl_2$, 0.5 mM $MnCl_2$, 25 mM KCl, 0.2 mM DTT, 40U Ribolock (Thermo Scientific) and 150 nM $\alpha-^{32}P$-ATP/UTP (Hartmann Analytic) for 2 h at 32 °C. Double-strand RNA was obtained by incubating RNAs 71 and 72 (Supplementary Table 4) at same molar concentration in annealing buffer (10 mM Tris pH 7.5, 50 mM KCl, 1 mM EDTA) at 95 °C for 5 min followed by incubation at RT for 1 h. RNA was extracted and detected as described above.

**sRNAs half-lives detection.** Wild type and cid14Δ cells with the ura4 hairpin expressed under the *nmt1* promoter were grown until log-phase. At time zero, the medium was supplemented with 15 µM thiamine to repress the *nmt1* promoter. Aliquots were taken after 1, 3 and 5 h to monitor the half-life of ura4 sRNAs. Total sRNAs and Ago1-bound sRNAs were purified and detected via Northern blotting analysis as described above. The quantity of the Ago1-bound sRNAs was determined by sybr gold staining of the 18% denaturing polyacrylamide gel. Equal amounts of sRNA were loaded on the gel. For quantification ura4 sRNAs from total fraction were normalized to snoR69 and Ago1-bound ura4 sRNAs were normalized to centromeric sRNAs. The experiments were performed at least in three biological replicates.

**Data availability.** The sequencing data that support the findings of this study have been deposited in the National Center for Biotechnology Information Gene Expression Omnibus (GEO) and are accessible through the GEO Series Accession Number GSE95821. All other relevant data are available from the corresponding author on request.

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

## Acknowledgements

We would like to thank François Bachand for kindly providing the Flag-Rrp6 plasmid. We thank Sigrun Jaklin for the excellent technical assistance and undergraduate student Henry Fabian Thomas for the assistance with molecular biology. We thank Silvija Bilokapic, Cornelia Brönner, Ilaria Ugolini and other members of the lab for the comments on the manuscript. This work was supported by the BioSysNet and ERC-smallRNAhet-309584.

## Author contributions

P.P. and M.H. designed the experiments. P.P. performed the experiments. P.P. and M.H. analysed the data. P.P. and M.H. wrote the paper.

## Additional information

**Competing interests:** The authors declare no competing financial interests.

