## [Peer review file · Nature Communications]

Reviewers' comments:

Reviewer #1 (Remarks to the Author):

This paper describes a potential role of non-canonical poly (A) polymerase Cid14 and poly(U) polymerase Cid16 in decay of Ago bound siRNA in fission yeas. This process may be important for preventing uncontrolled RNAi. Moreover, the Authors suggest that there is a feedback loop mechanism that silences genes essential for RNAi like Rdp1 in cells devoid of Cid14 and Cid16.

The presented hypothesis is appealing. Unfortunately, this paper suffers from several technical caveats, which strongly diminishes its scientific impact. Furthermore, results are often over-interpreted. The major novelty is that Ago-bound siRNAs are preferentially adenylated and uridylated in comparison to total siRNAs. Below I outline my major criticisms:

1) The Author claims that uridylation and adenylation induce siRNA decay. Although it is possible, there is actually no direct proof for that presented in the paper. The half-lives of analyzed siRNAs has not been measured. The quality of in vitro assay, in which the Authors mixed recombinant Cid14, Cid16 and Rrp6 with Ago-loaded siRNA, is so low that it is rather embarrassing to see it in the paper:

- The quality of gels is low and the final products are not well visible.
- There is no time course.
- There are no controls with catalytic mutants which would prove that the observed disappearance of the substrates is indeed dependent on the activity of the analyzed proteins.
- There are no replicates.

Furthermore, the RRP6 work in the context of the exosome complex and its activators like TRAMP. TRAMP consists of Cid14, an RNA binding protein, and RNA helicase. Thus the presented assay does not reflect the situation in vivo at all and because of that is not relevant for exosome mediated decay of Ago-bound siRNAs

2) Cid14 and RRP6 will influence the decay of nuclear RNA and by this as it was already established may influence the siRNA biogenesis. Thus without direct proof that Cid14 and Cid16 affect the half-lives of siRNAs the model presented in the paper is not sufficiently supported.

3) Rrp6 is a nuclear protein. How does cytoplasmic uridylation by Cid16 affect the decay, which presumably takes place in the nucleus? This should be further investigated. Are Ago associated uridylated siRNA imported to the nucleus? DIS3L2 seems to be a better candidate for cytoplasmic decay of uridylated siRNA. The authors should check the effect of DIS3L2 knockout on siRNA levels and their uridylation status.

4) The authors claim that Rdp1 is silenced post-transcriptionally because there is no increase in H3K9 methylation. This is just a suggestion that there is no epigenetic regulation while it is well possible that expression of Rdp1 is regulated at the level of transcription by specific transcription factors. Furthermore the levels of Rdp1 were measured by q-rtPCR and the RNA-seq results are not presented for this gene. Why it is so?

Reviewer #2 (Remarks to the Author):

Pisacane and Halic study the turnover of Argonaute bound small RNAs. In this process, the nucleotidyltransferases Cid14 and Cid16 are shown to adenylate and uridylate these RNAs, which can then be degraded by Rrp6. The authors suggest that this mechanism protects the genome from gene silencing by uncontrolled RNAi. The research topic is of interest and the experiments are well performed and mostly support the proposed model. However, the authors tend to make very strong statements that are not always fully supported by their results, and at times even contradictory.

Major comments:

1. Examples of strong statements that should be supported by further experiments or reasoning:

- a. page 9: "... and this is stably inherited". How do you know this?
- b. page 9: "...indicating that rRNA is efficiently silenced by RNAi". The rRNA reduction could be due to indirect effects e.g. due to cell stress. A more direct demonstration of its association with the RNAi machinery in the Rdp1 overexpressing cells would support the claim.

2. Other examples of overinterpretations:

- a. page 6: "these data show that only Argonaute-bound small RNAs are adenylated or uridylated..." This comes after having said that "centromeric siRNAs were modified at the same rate in both the total and the Argonaute-bound sample", as can be seen in Figure 1d.
- b. page 8: "Our data show that adenylation of small RNAs by Cid14 protects the genome from uncontrolled RNAi..."
- c. page 9: "Our data show that in *cid14Δ* and *cid14Δcid16Δ* cells, RNAi targets *rdp1* mRNA to protect the genome from uncontrolled RNAi..."

3. The claimed result from Figure 5b, especially for Cid14, is not clear from this autoradiograph. The authors claim it adds 1-2 adenines, which is hard to see.

4. If Rrp6 is involved in the degradation of sRNAs containing non-templated nucleotides at the 3'end, shouldn't these sRNAs accumulate in *rrp6Δ* strains? It's not what Figure 5c shows.

5. It would be relevant that the authors discuss their work in relation to reference 33, as similar kinds of analyses were performed there.

6. Contradictions that need some clarification:

- a. Cid16 is located in the cytoplasm and Rrp6 is nuclear. How do the authors envision the degradation of Cid16 modified RNAs by Rrp6?
- b. On page 7, when talking about IRC3, it is stated that the results indicate that uridylation is essential for RNA biogenesis. This is consistent with the findings in reference 33, but appears opposite to the model from Figure 6, where uridylation leads to decay. Add an

explanation about how in some cases the modification might participate in siRNA biogenesis.

c. On page 10, it is said that there's a population of RNAs where 5-10 uridines were added by Cid16. The next sentence claims that Cid14 and Cid16 add only 1-2 nucleotides.

Minor comments

1. When the authors refer to the RNA exosome, there's some confusion about how the core exosome and its associated exonucleases and co-factors interact and function, which should be revised:

a. abstract: "the nuclear exosome Rrp6". Rrp6 is an exonuclease that associates with the core exosome.

b. introduction, page 3: "3' exonuclease exosome". It should say at least "3' to 5'".

c. introduction, page 4: "TRAMP ... feeds these RNAs through the Rrp6 subunit to the core exosome". TRAMP is a cofactor of the exosome that hands transcripts to Rrp6 or the core exosome.

2. Figure 1e is referenced after a sentence talking about the whole genome, but it only shows a few examples. It would be better to provide some more global image of the data.

3. The legend text for Figures 1e and 1g talks about "centromeric region", but this is not what it represents.

4. Figure 3d does not show a very clear difference for several of the genes represented. A clearer way to show any different mRNA levels could be to show the actual rpkms values per gene or perform RT-qPCR assays.

5. In several figures, the symbol "/" is used on the y axis title of bar plots, but its meaning is not clear: in Figure 2d and supplementary Figure 2e, it seems to indicate that a fold change is calculated; in Figure 3b, it seems that it means "and"; it is not clear what it is in supplementary Figure 5d? Please explain.

6. On page 6, at the beginning of the second paragraph, the description of which RNAs are modified is confusing, as it says the same about centromeric RNAs twice (second and beginning of third sentence) and this is in contradiction with the first sentence.

7. The results section, regarding centromeric repeats and silencing, second paragraph on page 7, could contain a clearer description. In the first sentence "we observed only a smaller reduction...", smaller than what? The second last sentence on the page states "... is only moderately reduced", there could also be a comparison to the *clr4Δ* strain, so that it is clear what "moderately" refers to.

8. On page 8, when the authors talk about "the generation of siRNAs at many euchromatic genes...", it would be better to talk about "higher levels" or "higher expression", because, according to their model, the siRNAs are produced at the same rate, but are stabilised because of the lack of degradation.

9. On page 8, the sentence starting with "rRNA33..." is not precise enough, as it should say that siRNAs derived from mRNAs and ncRNAs accumulate.

10. On page 12, 4th line, in relation to a reference about uridylation by TUT4/7 "...confirming our result...". I think that "confirming" is not the best word here, as the other effect has been shown before; it would be better to use a different expression, such as: "in agreement with our result".

11. Western blotting and northern blotting analysis - not - 'western blot' or 'northern blot'.

Reviewer #3 (Remarks to the Author):

This manuscript shows that Cid14 and Cid16 adenylate and uridylate, respectively, small RNAs (sRNAs) in fission yeast, and interact with Argonaute to modify only Argonaute-associated sRNAs. Consistent with previous results, the authors observe small effects on the levels of centromeric siRNAs and centromeric silencing in *cid14Δ*, *cid16Δ* or *cid14Δ cid16Δ* double mutant cells, suggesting that the activities of Cid14/16 do not have major effects on centromeric siRNA function or biogenesis. However, the levels of Argonaute-bound sRNAs mapping to mRNAs or ncRNAs, which show higher levels of 3' modifications than other classes of sRNAs, are increased in *cid14Δ* and *cid14Δ cid16Δ* cells, suggesting that Cid14/16 is responsible for degradation or turnover of these sRNAs. The authors report that Cid14/16 recruit exosome component Rrp6 to Argonaute to degrade Argonaute-bound small RNAs. They further suggest that siRNAs mapping to genes/ncRNAs are functional and capable of mediating post-transcriptional silencing of their targets. Interestingly, *cid14Δ* and *cid14Δ rps6*; *cid16Δ* cells adapt to the higher levels of genome-wide sRNAs by post-transcriptionally silencing Rdp1 via RNAi, and this silencing of Rdp1 is important for viability of *cid14Δ* cells.

Overall, the data regarding the adenylation and uridylation activities of Cid14 and Cid16 are convincing, and illuminating with regards to sRNA metabolism and turnover. The results are important and provide new insight into RNA surveillance pathways and mechanisms of small RNA turnover. They should be of broad interest to the RNA silencing community. The authors should address the following minor concerns prior to publication.

1) The authors claimed that siRNAs generated at ectopic loci are fully functional and reduce the abundance of targeted transcripts, and presented selected examples of loci that showed increased levels of sRNAs and decreased transcript levels in *cid14Δ* or *cid14Δ cid16Δ* cells. The manuscript would be stronger if the authors could provide a more comprehensive genome-wide analysis of the correlation between siRNA levels and transcript levels, and/or qPCR analysis to confirm the small decreases in transcript levels that seem to be only based on a single RNA-seq experiment.

2) The claim that Cid14/16 recruit Rrp6 to degrade Argonaute-associated small RNAs is not based on direct evidence and can be made more convincing. Is there any type of control on sRNA level that the authors could include? Or another assay?

Dear reviewers,

We first would like to thank you for the time spent on our manuscript entitled "Cid14 and Cid16 nucleotidyltransferases promote degradation of Argonaute bound small RNAs by Rrp6 and protect the genome from uncontrolled RNAi" and for the valuable comments.

In the revised version we have addressed all comments and added set of additional experiments. Our study shows that Argonaute-bound small RNAs are actively removed from Argonaute and this is required for fidelity of RNAi. We show that nucleotidyltransferases Cid14/16 modify Argonaute-bound small RNAs and target them for degradation by nuclear exosome Rrp6. In the revised version we show that activity of Cid14/16 is required to recruit Rrp6 and degrade Argonaute-bound small RNAs. We also show that half-life of Argonaute-bound small RNAs is longer in *cid14* deletion cells which shows that Cid14 directly targets Argonaute-bound small RNAs for degradation. Failure in turnover of Argonaute-bound small RNAs results in accumulation of "noise" small RNAs on Argonaute and targeting of diverse euchromatic genes by RNAi in *cid14Δ* and *cid14Δcid16Δ* cells. To protect themselves from uncontrolled RNAi, these cells exploit the RNAi machinery to silence genes essential for RNAi itself, which is required for viability of *cid14Δ* cells. We observe that over-expression of Rdp1 in *cid14Δ* cells leads to targeting and silencing of rRNA and protein coding genes which reduces their viability.

Reviewers' comments:

Reviewer #1 (Remarks to the Author):

This paper describes a potential role of non-canonical poly (A) polymerase Cid14 and poly(U) polymerase Cid16 in decay of Ago bound siRNA in fission yeas. This process may be important for preventing uncontrolled RNAi. Moreover, the Authors suggest that there is a feedback loop mechanism that silences genes essential for RNAi like Rdp1 in cells devoid of Cid14 and Cid16.

The presented hypothesis is appealing. Unfortunately, this paper suffers from several technical caveats, which strongly diminishes its scientific impact. Furthermore, results are often over-interpreted. The major novelty is that Ago-bound siRNAs are preferentially adenylated and uridylated in comparison to total siRNAs. Below I outline my major criticisms:

1) The Author claims that uridylation and adenylation induce siRNA decay. Although it is possible, there is actually no direct proof for that presented in the paper. The half-lives of analyzed siRNAs has not been measured.

In the revised version of the manuscript we analyzed the half-lives of *ura4* siRNAs that are generated from a hairpin construct of *ura4* gene under the control of the repressible *nmt1* promoter. We expressed the *ura4* hairpin to generate *ura4* siRNA. After addition of thiamine to the media the *nmt1* promoter is repressed and new *ura4* siRNAs are generated at lower amounts. This allowed us to follow their degradation and half-life. We observed that siRNAs from the total fraction (Argonaute-bound and free small RNAs) are rapidly degraded in wild type cells. In *cid14Δ* cells after initial rapid degradation (free small RNAs), the degradation was slower (Argonaute-bound small RNAs).

We show that Argonaute-bound sRNAs have longer half life compared to the total fraction. In wild type cells we observe degradation of Argonaute-bound small RNAs, while in *cid14Δ* cells the degradation is impaired and the half life of Argonaute-bound small RNAs is prolonged (Figure 6).

The quality of in vitro assay, in which the Authors mixed recombinant Cid14, Cid16 and Rrp6 with Ago-loaded siRNA, is so low that it is rather embarrassing to see it in the paper:

- The quality of gels is low and the final products are not well visible.
- There is no time course.

We have repeated the experiments and included loading control. We also performed the time course experiment with both nucleotidyl-transferases in which degradation products are visible as shown in Figure 5h,i.

- There are no controls with catalytic mutants which would prove that the observed disappearance of the substrates is indeed dependent on the activity of the analyzed proteins.

We performed in vitro assays with the activity mutants of both nucleotidyl-transferases. We did not detect any nucleotidyl-transferase activity with Cid14DADA and Cid16DADA mutants on Argonaute-bound or free small RNA.

We also show that nucleotidyl-transferase activity of Cid14 and Cid16 is required for degradation of Ago1 bound small RNAs. We did not observe degradation of Argonaute-bound small RNAs when we used Cid14DADA and Cid16DADA mutants (Figure 5 f,g).

- There are no replicates.

We did at least 3 replicates for each experiment, we just did not add them to the manuscript. Biochemical experiments showing degradation are now shown in multiple assays (end point, time course, activity mutants).

Furthermore, the RRP6 work in the context of the exosome complex and its activators like TRAMP. TRAMP consists of Cid14, an RNA binding protein, and RNA helicase. Thus the presented assay does not reflect the situation in vivo at all and because of that is not relevant for exosome mediated decay of Ago-bound siRNAs

We agree that other subunits of TRAMP complex might promote Cid14 activity or Rrp6 recruitment, but they are clearly dispensable in a minimal system. Our data clearly show that Cid14 activity promotes degradation of Ago1-bound small RNAs and it is sufficient for degradation.

2) Cid14 and RRP6 will influence the decay of nuclear RNA and by this as it was already established may influence the siRNA biogenesis. Thus without direct proof that Cid14 and Cid16 affect the half-lives of siRNAs the model presented in the paper is not sufficiently supported.

We have determined the half-lives of Argonaute-bound and total small RNAs in wild type and *cid14* deletion cells (Figure 6). These data show that Cid14 is directly involved in degradation of Argonaute-bound small RNAs.

3) Rrp6 is a nuclear protein. How does cytoplasmic uridylation by Cid16 affect the decay, which presumably takes place in the nucleus? This should be further investigated. Are Ago associated uridylated siRNA imported to the nucleus? DIS3L2 seems to be a better candidate for cytoplasmic decay of uridylated siRNA. The authors should check the effect of DIS3L2 knockout on siRNA levels and their uridylation status.

It is not exactly clear how Argonaute is loaded with small RNAs, but some data indicate that Argonaute (in ARC complex) might shuttle from nucleus to cytoplasm in a process of small RNA loading/maturation. Argonaute is loaded with small RNAs in the ARC complex possibly in cytoplasm and imported into nucleus where Ago1 interacts with different set of proteins in RITS complex (Buker et al, 2007; Holoch and Moazed, 2015). It is possible that uridylated small RNAs are imported into nucleus and degraded by Rrp6. It is also possible that Cid16 is present also in the nucleus.

We did not observed a change in siRNA level or in the uridylation status of Ago1-bound small RNA in *dis3-54* or *dis3L2Δ* as shown in Figure 5c. This indicates that Dis3L2 and Dis3 are not involved in degradation of Ago1 bound small RNAs in fission yeast.

4) The authors claim that Rdp1 is silenced post-transcriptionally because there is no increase in H3K9 methylation. This is just a suggestion that there is no epigenetic regulation while it is well possible that expression of Rdp1 is regulated at the level of transcription by specific transcription factors. Furthermore the levels of Rdp1 were measured by q-rtPCR and the RNA-seq results are not presented for this gene. Why it is so?

We corrected our wording. Our data show that silencing is independent of heterochromatin. We analyzed RNA levels of *rdp1* by qPCR to have this data in 4 replicates (since this is an important finding) and to determine RNA levels in *cid14cid16* double mutant as well.

Reviewer #2 (Remarks to the Author):

Pisacane and Halic study the turnover of Argonaute bound small RNAs. In this process, the nucleotidyltransferases Cid14 and Cid16 are shown to adenylate and uridylate these RNAs, which can then be degraded by Rrp6. The authors suggest that this mechanism protects the genome from gene silencing by uncontrolled RNAi. The research topic is of interest and the experiments are well performed and mostly support the proposed model. However, the authors tend to make very strong statements that are not always fully supported by their results, and at times even contradictory.

Major comments:

1. Examples of strong statements that should be supported by further experiments or reasoning:

a. page 9: "... and this is stably inherited". How do you know this?

We corrected the phrase.

b. page 9: "...indicating that rRNA is efficiently silenced by RNAi". The rRNA reduction could be due to indirect effects e.g. due to cell stress. A more direct demonstration of its association with the RNAi machinery in the Rdp1 overexpressing cells would support the claim.

In revised version we sequenced small RNAs from Rdp1 over-expression strains (see Figure 4d) and observed strong increase of small RNAs that are generated from rRNA. Now we find siRNAs over the whole rRNA and not only at the 3'ETS and end of 25S. This shows that RNAi directly targets rRNA in cells that over-express Rdp1. Moreover, ChIP-seq experiments showed an enrichment of H3K9me2 at the rDNA locus (Figure S4f). This indicates that small RNAs observed in the Rdp1 over-expression do induce higher levels of H3K9 methylation and heterochromatin formation at rDNA.

2. Other examples of overinterpretations:

a. page 6: "these data show that only Argonaute-bound small RNAs are adenylated or uridylated..." This comes after having said that "centromeric siRNAs were modified at the same rate in both the total and the Argonaute-bound sample", as can be seen in Figure 1d.

Majority of mRNA or rRNA degradation products found in total fraction are not loaded on Argonaute and they are not adenylated or uridylated. Centromeric siRNA are modified at similar rate suggesting that majority of centromeric siRNAs are indeed loaded on Argonaute.

We write this more carefully.

b. page 8: "Our data show that adenylation of small RNAs by Cid14 protects the genome from uncontrolled RNAi..."

In *cid14* deletion cells we observe high rate of RNAi at protein coding genes which are normally not targeted by the RNAi machinery in wild type cells. This suggests that Cid14/Rrp6 mediated degradation of Ago1-bound small RNAs protects the genome from uncontrolled RNAi. In Rdp1 overexpression even more genes are targeted by RNAi.

We write this more carefully.

c. page 9: "Our data show that in *cid14Δ* and *cid14Δcid16Δ* cells, RNAi targets *rdp1* mRNA to protect the genome from uncontrolled RNAi..."

We observe high rate of RNAi at protein coding genes in *cid14* deletion cells which suggests that Cid14 mediated degradation of Ago1-bound small RNAs protects the genome from RNAi. By “uncontrolled RNAi” we wanted to point out the fact that in *cid14Δ* and *cid14Δcid16Δ* cells we observed small RNA generation at genes which are not targets of RNAi machinery in wild type cells. The RNAi in this cells is targeting random protein coding genes.

When we over-expressed *rdp1* we observed an increase in siRNA generation at rDNA and protein coding genes. This shows that silencing of *rdp1* is essential to reduce this uncontrolled RNAi that targets mRNAs and rRNA.

We write this more carefully.

3. The claimed result from Figure 5b, especially for Cid14, is not clear from this autoradiograph. The authors claim it adds 1-2 adenines, which is hard to see.

We repeated the adenylation assays and included the activity mutants. For Cid16 we observed only a single product migrating just above 22 nucleotides indicating that Cid16 adds ~1-3 nucleotides to Argonaute-bound small RNA. For Cid14 we observed two products suggesting that Cid14 adds either ~1-3 nucleotides (band migrating around 22 nucleotide marker) or an oligo-A tail (band migrating above 30 nucleotide marker).

4. If Rrp6 is involved in the degradation of sRNAs containing non-templated nucleotides at the 3'end, shouldn't these sRNAs accumulate in *rrp6Δ* strains? It's not what Figure 5c shows.

We expected this as well. However, non templated adenines at the 3' end will be removed by Triman nuclease (PARN family deadenylase). Addition of adenines recruits both Triman and Rrp6. Triman is however not capable to remove small RNAs from Argonaute (Figure 5d,e; Marasovic et al, 2013). Triman will trim small RNA to 22nt and effectively remove untemplated adenines and make small RNA functional again. We have previously shown that longer small RNAs (above 24 nt) are not functional in slicing complementary targets.

5. It would be relevant that the authors discuss their work in relation to reference 33, as similar kinds of analyses were performed there.

We added more discussion related to reference 33. Although they have studied RNAi in *cid14* deletion cells, most of our findings do not overlap. In Bühler *et al* authors observed ectopic RNAi at rRNA, however depth of the sequencing was too low to observe ectopic RNAi at other loci. They did not analyze addition of non-templated nucleotides by Cid14 nor degradation of small RNAs.

6. Contradictions that need some clarification:

a. Cid16 is located in the cytoplasm and Rrp6 is nuclear. How do the authors envision the degradation of Cid16 modified RNAs by Rrp6?

First, we cannot exclude that Cid16 is also in the nucleus. Second, it has been proposed that Argonaute is loaded with small RNAs in ARC complex and imported into nucleus where Ago1 interacts with different set of proteins in RITS complex. It is possible that uridylylated small RNAs are

imported into nucleus and degraded by Rrp6 in nucleus. Alternatively, Cid16 could also be in the nucleus.

We did not observe a change in siRNA level or in the uridylation status of Ago1-bound small RNA in *dis3-54* or *dis3L2Δ* cells as shown in Figure 5c. This indicates that *dis3L2* and *dis3* are not involved in degradation of Ago1 bound small RNAs.

b. On page 7, when talking about IRC3, it is stated that the results indicate that uridylation is essential for RNA biogenesis. This is consistent with the findings in reference 33, but appears opposite to the model from Figure 6, where uridylation leads to decay. Add an explanation about how in some cases the modification might participate in siRNA biogenesis.

We corrected our wording. We observe that siRNAs at IRC3 are reduced in *cid16/14* deletion cells. However, we actually do not think that uridylation contributes directly to biogenesis of IRC3. We think that loss of siRNAs in *cid16* deletion cells is indirect. IRC3 siRNAs are reduced in most mutants we have studied (Halic and Moazed, 2010; Marasovic et al, 2013). It seems that at this element siRNA biogenesis is particularly sensitive to any perturbation.

c. On page 10, it is said that there's a population of RNAs where 5-10 uridines were added by Cid16. The next sentence claims that Cid14 and Cid16 add only 1-2 nucleotides.

We performed again the nucleotidyl-transferase assays, including the activity mutants. For Cid16 we observe only a product around 23-25 nt, as shown in Figure 5b. This indicates that Cid16 adds 1-3 nucleotides to Argonaute-bound small RNAs. For Cid14 we observe 2 products. First product is migrating around 23-25 nucleotides, while second product migrates above 30 nucleotides. This indicates that Cid14 can add either 1-3 nucleotides or > 10 nucleotides to Argonaute-bound small RNAs.

Minor comments

We improved the text according to the suggestions from Reviewer #2.

1. When the authors refer to the RNA exosome, there's some confusion about how the core exosome and its associated exonucleases and co-factors interact and function, which should be revised:

a. abstract: "the nuclear exosome Rrp6". Rrp6 is an exonuclease that associates with the core exosome.

We changed the sentence with "3'-5' exonuclease Rrp6"

b. introduction, page 3: "3' exonuclease exosome". It should say at least "3' to 5'".

We corrected this.

c. introduction, page 4: "TRAMP ... feeds these RNAs through the Rrp6 subunit to the core exosome". TRAMP is a cofactor of the exosome that hands transcripts to Rrp6 or the core exosome.

We corrected this.

2. Figure 1e is referenced after a sentence talking about the whole genome, but it only shows a few examples. It would be better to provide some more global image of the data.

We added pie charts showing distribution of uniquely mapped small RNA (Figure S1g) in wild type cells. The pie chart shows that small RNAs having U or A at the 3' end map to all genomic regions. Small RNAs that have U at 3' end are enriched for priRNAs that originate from mRNAs. Small RNAs with A at 3' end are enriched for priRNAs originating from ncRNAs.

3. The legend text for Figures 1e and 1g talks about "centromeric region", but this is not what it represents.

We corrected this.

4. Figure 3d does not show a very clear difference for several of the genes represented. A clearer way to show any different mRNA levels could be to show the actual rpkm values per gene or perform RT-qPCR assays.

We show the actual rpkm values in Supplementary Figure 3e. In figure 3e we now show change in mRNA level for all genes that generate siRNAs in *cid14* deletion cells. These genes show reduced levels of mRNA in *cid14* deletion cells.

5. In several figures, the symbol "/" is used on the y axis title of bar plots, but its meaning is not clear: in Figure 2d and supplementary Figure 2e, it seems to indicate that a fold change is calculated; in Figure 3b, it seems that it means "and"; it is not clear what it is in supplementary Figure 5d? Please explain.

We have corrected this. The symbol "/" always means a fold change.

6. On page 6, at the beginning of the second paragraph, the description of which RNAs are modified is confusing, as it says the same about centromeric RNAs twice (second and beginning of third sentence) and this is in contradiction with the first sentence.

We improved the text in this part of the manuscript. We hope the description is clear now.

7. The results section, regarding centromeric repeats and silencing, second paragraph on page 7, could contain a clearer description. In the first sentence "we observed only a smaller reduction...", smaller than what? The second last sentence on the page states "... is only

moderately reduced", there could also be a comparison to the *clr4Δ* strain, so that it is clear what "moderately" refers to.

We included the data of *dcr1* deletion strain which shows a complete derepression of centromeric silencing. When compared to *dcr1* deletion, the loss of silencing in *cid14/16* deletion strain is small.

8. On page 8, when the authors talk about "the generation of siRNAs at many euchromatic genes...", it would be better to talk about "higher levels" or "higher expression", because, according to their model, the siRNAs are produced at the same rate, but are stabilised because of the lack of degradation.

Our data show that siRNAs are generated at euchromatic genes only in mutant strains. The wild type does not generate detectable siRNA at these genes. Low levels of primary priRNAs can be detected (Halic and Moazed, 2010; Marasovic et al, 2013), but they are probably rapidly removed before they initiate generation of secondary siRNAs. According to our model, siRNAs are generated at these genes because of stabilization of priRNAs or very low abundant siRNAs. This stabilization of priRNAs starts the amplification loop to make more siRNAs.

According to our data siRNAs are not produced at these loci, but their production is initiated by stabilization of primary priRNAs.

9. On page 8, the sentence starting with "rRNA33..." is not precise enough, as it should say that siRNAs derived from mRNAs and ncRNAs accumulate.

We changed the sentence.

10. On page 12, 4th line, in relation to a reference about uridylation by TUT4/7 "...confirming our result...". I think that "confirming" is not the best word here, as the other effect has been shown before; it would be better to use a different expression, such as: "in agreement with our result".

We changed it.

11. Western blotting and northern blotting analysis - not - 'western blot' or 'northern blot'.

We corrected this.

Reviewer #3 (Remarks to the Author):

This manuscript shows that Cid14 and Cid16 adenylate and uridylate, respectively, small RNAs (sRNAs) in fission yeast, and interact with Argonaute to modify only Argonaute-associated sRNAs. Consistent with previous results, the authors observe small effects on the levels of centromeric siRNAs and centromeric silencing in *cid14Δ*, *cid16Δ* or *cid14Δ cid16Δ* double mutant cells, suggesting that the activities of Cid14/16 do not have major effects on centromeric siRNA function or biogenesis. However, the levels of Argonaute-bound sRNAs mapping to mRNAs or ncRNAs, which show higher levels of 3' modifications than other

classes of sRNAs, are increased in *cid14Δ* and *cid14Δ cid16Δ* cells, suggesting that Cid14/16 is responsible for degradation or turnover of these sRNAs. The authors report that Cid14/16 recruit exosome component Rrp6 to Argonaute to degrade Argonaute-bound small RNAs. They further suggest that siRNAs mapping to genes/ncRNAs are functional and capable of mediating post-transcriptional silencing of their targets. Interestingly, *cid14Δ* and *cid14Δ cid16Δ* cells adapt to the higher levels of genome-wide sRNAs by post-transcriptionally silencing Rdp1 via RNAi, and this silencing of Rdp1 is important for viability of *cid14Δ* cells.

Overall, the data regarding the adenylation and uridylation activities of Cid14 and Cid16 convincing, and illuminating with regards to sRNA metabolism and turnover. The results are important and provide new insight into RNA surveillance pathways and mechanisms of small RNA turnover. They should be of broad interest to the RNA silencing community. The authors should address the following minor concerns prior to publication.

1) The authors claimed that siRNAs generated at ectopic loci are fully functional and reduce the abundance of targeted transcripts, and presented selected examples of loci that showed increased levels of sRNAs and decreased transcript levels in *cid14Δ* or *cid14Δ cid16Δ* cells. The manuscript would be stronger if the authors could provide a more comprehensive genome-wide analysis of the correlation between siRNA levels and transcript levels, and/or qPCR analysis to confirm the small decreases in transcript levels that seem to be only based on a single RNA-seq experiment.

As suggested by the reviewer we performed genome wide analysis. We show now that genes that generate siRNAs have reduced RNA levels in *cid14* deletions cell (Figure 3e). We performed qPCR experiment for *rdp1*.

2) The claim that Cid14/16 recruit Rrp6 to degrade Argonaute-associated small RNAs is not based on direct evidence and can be made more convincing. Is there any type of control on sRNA level that the authors could include? Or another assay?

We included a 30 nt long DNA loading control. We have also included time course experiments showing the presence of intermediate degradation products. We also show that half-life of Argonaute-bound small RNAs is longer in *cid14* deletion cells than in wild type cells (Figure 6).

Reviewers' comments:

Reviewer #1 (Remarks to the Author):

The revised version manuscript is improved but several unresolved issues remain and presented data does not sufficiently support the major claims:

1)The major issue was the influence of tailing on the half-lives of siRNAs in vivo. The Authors measured the half-lives of ura4 siRNAs that are generated from a hairpin construct of ura4 gene under the control of the repressible nmt1 promoter. They postulate that “Argonaute-bound siRNAs have longer half-life compared to the total fraction.” Unfortunately the putative “siRNAs” seen in northern blots in the total fractions do not migrate as a sharp band. Moreover, in cid14Δ cells this smearing is even more pronounced. Thus, it is well possible that degradation intermediates rather than siRNAs are observed. Such intermediates actually accumulate in cid14Δ cells, which is not surprising knowing that cid14 as a part of the TRAMP complex is involved in the nuclear RNA surveillance. However, the whole gel should be presented to be able to judge it. The whole gel is also essential to see the behavior and stability of the full length reporter.

The Authors further claim “In wild type cells we observe degradation of Argonaute-bound small RNAs, while in cid14Δ cells the degradation is impaired and the half-life of Argonaute-bound small RNAs is prolonged”, which may be correct. However, since AGO-bound siRNAs are far more stable than the free ones it is expected that in longer time points we should observe the stabilization of total siRNA in cid14Δ cells, which is not the case. Thus it is well possible that new siRNAs are generated from the degradation intermediates accumulating in the cid14Δ cells during the chase period.

2)Another important concern was related to in vitro assays. They are indeed improved but the experiments with the entire exosome would be more relevant. Honestly, I suspect that uridylation or adenylation of Ago-bound siRNAs can enhance the decay by essentially every exoribonuclease. Without solid proof that indeed tailing enhance decay of siRNA in vivo (see above) such an assay does not sufficiently support the major claims of the paper

3)The question how does cytoplasmic uridylation by Cid16 affect the decay, which presumably takes place in the nucleus, also remains unanswered. The author ruled out the involvement of DIS3L2 only. Surprisingly, data are shown in figure 5C but they are not mentioned in the revised manuscript. The data on dis3-54 are not shown at all.

4)Finally one of very important issues was the mechanism of Rdp1 silencing. During the first round of the review process I asked why the levels of Rdp1 were measured by q-rtPCR and RNA-seq results are not presented for this gene. This question remained unanswered. Moreover, the transcriptional regulation of Rdp1 was not ruled out by the Authors.

Reviewer #2 (Remarks to the Author):

The authors have addressed some of my concerns. However, most were done carelessly and therefore the paper still contains over-interpretations and contradictory statements, which at times show a lack of scientific rigor. Below I include comments to the answers of the authors that do not adequately address my concerns.

Reviewer #2 (Remarks to the Author):

2. Other examples of overinterpretations:

a. page 6: "these data show that only Argonaute-bound small RNAs are adenylated or uridylated..." This comes after having said that "centromeric siRNAs were modified at the same rate in both the total and the Argonaute-bound sample", as can be seen in Figure 1d. Majority of mRNA or rRNA degradation products found in total fraction are not loaded on Argonaute and they are not adenylated or uridylated. Centromeric siRNA are modified at similar rate suggesting that majority of centromeric siRNAs are indeed loaded on Argonaute.

We write this more carefully.

My comment: The word "suggesting" used above would be better to include in the text; rather than using the word "show". This is because you mainly have a correlation of centromeric siRNAs being more loaded onto Argonaute and more modified while the rest of siRNAs being only a small fraction of the loaded RNAs and rarely modified in the total fraction. The sentence with which I started this comment has unfortunately not been changed in the manuscript.

b. page 8: "Our data show that adenylation of small RNAs by Cid14 protects the genome from uncontrolled RNAi..."

In cid14 deletion cells we observe high rate of RNAi at protein coding genes which are normally not targeted by the RNAi machinery in wild type cells. This suggests that Cid14/Rrp6 mediated degradation of Ago1-bound small RNAs protects the genome from uncontrolled RNAi. In Rdp1 overexpression even more genes are targeted by RNAi.

We write this more carefully.

My comment: In this response, the authors use the verb "suggests", which again would be appropriate for the manuscript text as well. The sentence in the manuscript has not been changed; the only thing is that in the next sentence the word "turnover" has been changed to "surveillance".

c. page 9: "Our data show that in cid14 Δ and cid14 Δ cid16 Δ cells, RNAi targets rdp1 mRNA to protect the genome from uncontrolled RNAi..."

We observe high rate of RNAi at protein coding genes in cid14 deletion cells which suggests that Cid14 mediated degradation of Ago1-bound small RNAs protects the genome from RNAi. By "uncontrolled RNAi" we wanted to point out the fact that in cid14 Δ and

cid14Δcid16Δ cells we observed small RNA generation at genes which are not targets of RNAi machinery in wild type cells. The RNAi in this cells is targeting random protein coding genes.

When we over-expressed rdp1 we observed an increase in siRNA generation at rDNA and protein coding genes. This shows that silencing of rdp1 is essential to reduce this uncontrolled RNAi that targets mRNAs and rRNA.

We write this more carefully.

This is again written in the same way, except for the removal of “stably inherited” in response to my first comment. I agree with the authors that all these ideas are plausible and in agreement with the results, but the sentence is an overstatement, as rdp1 might be targeted randomly, as many other genes and, as a consequence, it contributes to dampen the effects of the excessive RNAi.

5. It would be relevant that the authors discuss their work in relation to reference 33, as similar kinds of analyses were performed there.

We added more discussion related to reference 33. Although they have studied RNAi in cid14 deletion cells, most of our findings do not overlap. In Bühler et al authors observed ectopic RNAi at rRNA, however depth of the sequencing was too low to observe ectopic RNAi at other loci. They did not analyze addition of non-templated nucleotides by Cid14 nor degradation of small RNAs.

My comment: Because they studied RNAi in cid14 deletion cells and the findings do not overlap, it would be really interesting to discuss the reasons why this is the case, which might be difference in depth or any other experimental difference. The reference in the new version is number 34 and, as far as I can see, there is now an additional sentence that refers to it, which is not really discussing more about their work.

c. On page 10, it is said that there's a population of RNAs where 5-10 uridines were added by Cid16. The next sentence claims that Cid14 and Cid16 add only 1-2 nucleotides.

We performed again the nucleotidyl-transferase assays, including the activity mutants. For Cid16 we observe only a product around 23-25 nt, as shown in Figure 5b. This indicates that Cid16 adds 1-3 nucleotides to Argonaute-bound small RNAs. For Cid14 we observe 2 products. First product is migrating around 23-25 nucleotides, while second product migrates above 30 nucleotides. This indicates that Cid14 can add either 1-3 nucleotides or > 10 nucleotides to Argonaute-bound small RNAs.

My comment: It is a bit uncomfortable that you obtain the opposite result than previously, in the sense that Cid16 now gives rise to one modified population and Cid14 to two. How reproducible is this? But still, the text is not precise enough, as you say: “similarly to Cid14, Cid16 added mostly 1-2 nucleotides...” This is after you have said: “Cid14 added either 1-3 or 15-20 non-templated adenines...” And when one looks at the gel it does not seem that the population with fewer added nucleotides is larger than the other one.

Minor comments

We improved the text according to the suggestions from Reviewer #2.

1. When the authors refer to the RNA exosome, there's some confusion about how the core

exosome and its associated exonucleases and co-factors interact and function, which should be revised:

a. abstract: "the nuclear exosome Rrp6". Rrp6 is an exonuclease that associates with the core exosome.

We changed the sentence with "3'-5' exonuclease Rrp6".

My comment: It is changed in the abstract, but still wrong in the discussion and even in the rebuttal letter.

7. The results section, regarding centromeric repeats and silencing, second paragraph on page 7, could contain a clearer description. In the first sentence "we observed only a smaller reduction...", smaller than what? The second last sentence on the page states "... is only moderately reduced", there could also be a comparison to the *clr4Δ* strain, so that it is clear what "moderately" refers to.

We included the data of *dcr1* deletion strain which shows a complete derepression of centromeric silencing. When compared to *dcr1* deletion, the loss of silencing in *cid14/16* deletion strain is small.

My comment: It is good to include the *dcr1* deletion for comparison. Regarding the first sentence, at least now it is clear that it is small in relation to wild-type cells, but I am not sure to what extent a reduction from 85% to 57 % can be considered small.

Reviewer #3 (Remarks to the Author):

The authors have successfully addressed my concerns regarding genome-wide analysis of the activity of euchromatic siRNAs and *Cid14/16*-based degradation of Argonaute-associated sRNAs. I recommend publication.

Reviewers' comments:

Reviewer #1 (Remarks to the Author):

The revised version manuscript is improved but several unresolved issues remain and presented data does not sufficiently support the major claims:

1)The major issue was the influence of tailing on the half-lives of siRNAs in vivo. The Authors measured the half-lives of *ura4* siRNAs that are generated from a hairpin construct of *ura4* gene under the control of the repressible *nmt1* promoter. They postulate that “Argonaute-bound siRNAs have longer half-life compared to the total fraction.” Unfortunately the putative “siRNAs” seen in northern blots in the total fractions do not migrate as a sharp band. Moreover, in *cid14*Δ cells this smearing is even more pronounced. Thus, it is well possible that degradation intermediates rather than siRNAs are observed. Such intermediates actually accumulate in *cid14*Δ cells, which is not surprising knowing that *cid14* as a part of the TRAMP complex is involved in the nuclear RNA surveillance. However, the whole gel should be presented to be able to judge it. The whole gel is also essential to see the behavior and stability of the full length reporter.

In fission yeast total siRNAs show smearing appearance since Dicer is not generating siRNAs of defined size (Marasovic et al, 2013; Colmenares et al, 2007). siRNAs are trimmed to the final length by Triman nuclease after they bind Argonaute (Marasovic et al, 2013). In Shimada et al., 2016, the authors sequenced the siRNAs deriving from the same *ura4* hairpin construct, showing that their length ranges from 19 to 25 nt.

We now show the whole gel of the northern (Figure 6a). We observe several higher products that are detected by our probes, but their amount is similar in wt and *cid14* deletion cells. This indicates that Cid14 is not involved in degradation of the double stranded hairpin. The full length reporter is not visible on the northern since we size selected RNA that are smaller than 200bp before loading the gel. But, we show by qPCR (Supplementary fig. 6b) that the full length construct is not more stable in *cid14* deletion cells. The construct is dsRNA and it is likely not degraded by the exosome. We observe

that construct is present at 10-15% of maximum level after 5h which will also lead to generation of new siRNA, but at the similar rate in both wild type and *cid14* deletion cells.

To confirm that these are indeed siRNA we deleted Dicer in these strains. *dcr1* and *cid14dcr1* deletion cells show that observed *ura4* small RNAs are Dicer products and therefore genuine siRNAs (Supplementary Fig. 6c). In *dcr1* deletion cells we do not observe small RNAs migrating between 20 and 30 nucleotides showing that these are genuine siRNAs and not degradation products.

The Authors further claim "In wild type cells we observe degradation of Argonaute-bound small RNAs, while in *cid14Δ* cells the degradation is impaired and the half-life of Argonaute-bound small RNAs is prolonged", which may be correct. However, since AGO-bound siRNAs are far more stable than the free ones it is expected that in longer time points we should observe the stabilization of total siRNA in *cid14Δ* cells, which is not the case. Thus it is well possible that new siRNAs are generated from the degradation intermediates accumulating in the *cid14Δ* cells during the chase period.

We do observe stabilization of total siRNAs in *cid14* deletion cells as indicated by the reviewer (Figure 6a,b). We also observe stabilization of Argonaute-bound small RNAs in *cid14* deletion cells (Figure 6c, d). We have to mention that we do not know what fraction of *ura4* small RNAs is loaded onto Argonaute. At centromeric repeats small RNA generation is coupled to H3K9me which recruits Ago1 to chromatin (through Chp1 subunit of RITS complex). In previous work H3K9me was not observed at *ura4* hairpin construct (Iida et al, 2008). This will reduce Argonaute localization to the locus and loading of siRNAs. Substantial fraction of *ura4* siRNAs is most likely not loaded on Argonaute and is degraded by Eri1 nuclease before loading (Iida et al, 2007). In *cid14* deletion cells we observe stabilization of Argonaute bound small RNAs and total siRNAs at later time point (which are likely Argonaute-bound). We included qPCR showing that full length precursor is not more stable in *cid14* deletion cells indicating that at later time points new siRNAs are not generated at higher rate in *cid14* deletion cells.

2) Another important concern was related to *in vitro* assays. They are indeed improved but the experiments with the entire exosome would be more relevant. Honestly, I suspect that uridylation or adenylation of Ago-bound siRNAs can enhance the decay by essentially every exonuclease. Without solid proof that indeed tailing enhances decay of siRNA *in vivo* (see above) such an assay does not sufficiently support the major claims of the paper

We actually observed that uridylation and adenylation do not enhance decay by every nuclease. For example nuclease Triman was not able to degrade Argonaute bound small RNA, although it trims them to 22 nucleotide length (Marasovic et al, 2013). Also adenylation or uridylation by Cid14/16 did not lead to degradation of Argonaute-bound small RNAs by Triman nuclease (Figure 5d,e and Marasovic et al, 2013). These data show that not every nuclease can degrade Argonaute-bound small RNAs in *in vitro* assays.

3) The question how does cytoplasmic uridylation by Cid16 affect the decay, which presumably takes place in the nucleus, also remains unanswered. The author ruled out the involvement of DIS3L2 only. Surprisingly, data are shown in figure 5C but they are not mentioned in the revised manuscript. The data on dis3-54 are not shown at all.

We included now the data for both Dis3-54 (already published earlier) and Dis3L2. We mentioned them now in the text.

As mentioned in the revised version, the Cid16 localization could be cytoplasmic as observed in genome wide localization study by Matsuyama, A et al, 2006, Nat. Biotechnol. It is also possible that fraction of Cid16 is localized in nucleus. We find that priRNAs originating from mRNAs are more often uridylated and ncRNA are more often adenylated which would support distinct localization of Cid16

and Cid14. Previous studies in fission yeast suggested that small RNAs might be loaded in cytoplasm. It was suggested that Arb complex could be involved in loading and shuttling of Argonaute (Holoch et al, 2015; Buker et al, 2007). It is also possible that mRNA degradation products are loaded on Argonaute during import to nucleus (after synthesis) and are removed by exosome in nucleus where Argonaute is predominantly localized in fission yeast. We show that decay is dependent on Cid16 activity and it is possible that uridylation recruits exosome after Argonaute is imported to nucleus. The reviewer should also consider that cytoplasmic RNAi was never shown in fission yeast. All current data indicate that only nuclear RNAi pathway exists in fission yeast. We think, that detailed analysis of potential cytoplasmic RNAi, cytoplasmic small RNA loading and small RNA labeling for degradation in cytoplasm is a subject for another study.

4) Finally one of very important issues was the mechanism of Rdp1 silencing. During the first round of the review process I asked why the levels of Rdp1 were measured by q-rtPCR and RNA-seq results are not presented for this gene. This question remained unanswered. Moreover, the transcriptional regulation of Rdp1 was not ruled out by the Authors.

As already mentioned we analyzed Rdp1 mRNA level by RT-qPCR since we focus on this single gene and wanted to confirm RNA sequencing data. We also wanted to determine Rdp1 mRNA level in *cid14cid16* double mutant. We have now included Rdp1 mRNA level from RNAseq in Supplementary Fig. 3i.

Our data show that *rdp1* gene is not regulated by heterochromatin. It is possible that some other way of transcriptional regulation is involved, but in our opinion this is out of scope of this study. We think that *rdp1* mRNA (and all other silenced RNA) are degraded post-transcriptionally. We do not think *rdp1* is specifically targeted. We think that *rdp1* was targeted randomly as any other mRNA, but cells that silence *rdp1* are growing much faster and will be selected. These cells are the fittest survivors of uncontrolled RNAi (Figure 4).

Reviewer #2 (Remarks to the Author):

The authors have addressed some of my concerns. However, most were done carelessly and therefore the paper still contains over-interpretations and contradictory statements, which at times show a lack of scientific rigor. Below I include comments to the answers of the authors that do not adequately address my concerns.

Reviewer #2 (Remarks to the Author):

We apologize to the reviewer for being careless in writing. We have fixed now all the over-interpretations raised by the reviewer.

2. Other examples of overinterpretations:

a. page 6: "these data show that only Argonaute-bound small RNAs are adenylated or uridylated..." This comes after having said that "centromeric siRNAs were modified at the same rate in both the total and the Argonaute-bound sample", as can be seen in Figure 1d.

Majority of mRNA or rRNA degradation products found in total fraction are not loaded on Argonaute and they are not adenylated or uridylated. Centromeric siRNA are modified at similar rate suggesting that majority of centromeric siRNAs are indeed loaded on Argonaute.

We write this more carefully.

My comment: The word "suggesting" used above would be better to include in the text; rather than using the word "show". This is because you mainly have a correlation of centromeric siRNAs being

more loaded onto Argonaute and more modified while the rest of siRNAs being only a small fraction of the loaded RNAs and rarely modified in the total fraction. The sentence with which I started this comment has unfortunately not been changed in the manuscript.

We have changed the word “show” to “suggest”.

b. page 8: "Our data show that adenylation of small RNAs by Cid14 protects the genome from uncontrolled RNAi..."

In *cid14* deletion cells we observe high rate of RNAi at protein coding genes which are normally not targeted by the RNAi machinery in wild type cells. This suggests that Cid14/Rrp6 mediated degradation of Ago1-bound small RNAs protects the genome from uncontrolled RNAi. In Rdp1 overexpression even more genes are targeted by RNAi.

We write this more carefully.

My comment: In this response, the authors use the verb “suggests”, which again would be appropriate for the manuscript text as well. The sentence in the manuscript has not been changed; the only thing is that in the next sentence the word “turnover” has been changed to “surveillance”.

We have changed the word “show” to “suggest”.

c. page 9: "Our data show that in *cid14Δ* and *cid14Δcid16Δ* cells, RNAi targets *rdp1* mRNA to protect the genome from uncontrolled RNAi..."

We observe high rate of RNAi at protein coding genes in *cid14* deletion cells which suggests that Cid14 mediated degradation of Ago1-bound small RNAs protects the genome from RNAi. By “uncontrolled RNAi” we wanted to point out the fact that in *cid14Δ* and *cid14Δcid16Δ* cells we observed small RNA generation at genes which are not targets of RNAi machinery in wild type cells. The RNAi in this cells is targeting random protein coding genes.

When we over-expressed *rdp1* we observed an increase in siRNA generation at rDNA and protein coding genes. This shows that silencing of *rdp1* is essential to reduce this uncontrolled RNAi that targets mRNAs and rRNA.

We write this more carefully.

This is again written in the same way, except for the removal of “stably inherited” in response to my first comment. I agree with the authors that all these ideas are plausible and in agreement with the results, but the sentence is an overstatement, as *rdp1* might be targeted randomly, as many other genes and, as a consequence, it contributes to dampen the effects of the excessive RNAi.

Actually we think that *rdp1* is targeted randomly and the cell that targeted *rdp1* and suppressed excessive RNAi grow faster and eventually dominated the culture. We think that the fastest growing cell was eventually selected. We added this in the discussion. However, detailed analysis is required to confirm this.

But this does not change the fact that RNAi targets *rdp1* in our cells (either randomly or possibly even in a targeted way) and this protects the genome from an excessive RNAi.

But, we changed the word “show” to “suggest”.

5. It would be relevant that the authors discuss their work in relation to reference 33, as similar kinds of analyses were performed there.

We added more discussion related to reference 33. Although they have studied RNAi in *cid14* deletion cells, most of our findings do not overlap. In Bühler et al authors observed ectopic RNAi at rRNA,

however depth of the sequencing was too low to observe ectopic RNAi at other loci. They did not analyze addition of non-templated nucleotides by Cid14 nor degradation of small RNAs.

My comment: Because they studied RNAi in cid14 deletion cells and the findings do not overlap, it would be really interesting to discuss the reasons why this is the case, which might be difference in depth or any other experimental difference. The reference in the new version is number 34 and, as far as I can see, there is now an additional sentence that refers to it, which is not really discussing more about their work.

One major difference is dept of the small RNA sequencing data. In the previous study less than 100k reads were obtained and many reads were contaminating rRNA and tRNA products. We sequenced >10M reads and our small RNA sequencing data are much cleaner (>98% 5'U). This allowed us to detect all this euchromatic small RNAs. Also our study focuses on small RNA modification and small RNA degradation. In previous study authors did not focus on small RNAs.

c. On page 10, it is said that there's a population of RNAs where 5-10 uridines were added by Cid16. The next sentence claims that Cid14 and Cid16 add only 1-2 nucleotides. We performed again the nucleotidyl-transferase assays, including the activity mutants. For Cid16 we observe only a product around 23-25 nt, as shown in Figure 5b. This indicates that Cid16 adds 1-3 nucleotides to Argonaute-bound small RNAs. For Cid14 we observe 2 products. First product is migrating around 23-25 nucleotides, while second product migrates above 30 nucleotides. This indicates that Cid14 can add either 1-3 nucleotides or > 10 nucleotides to Argonaute-bound small RNAs.

My comment: It is a bit uncomfortable that you obtain the opposite result than previously, in the sense that Cid16 now gives rise to one modified population and Cid14 to two. How reproducible is this? But still, the text is not precise enough, as you say: "similarly to Cid14, Cid16 added mostly 1-2 nucleotides..." This is after you have said: "Cid14 added either 1-3 or 15-20 non-templated adenines..." And when one looks at the gel it does not seem that the population with fewer added nucleotides is larger than the other one.

Both assay were repeated at least 5 times. With Cid14 we get often both bands, but the intensity of the higher band varies. Although the intensity of upper band looks similar to the lower band, one has to consider that upper band has 15-20 radioactive A incorporated and lower band has only 1-2. This means that the signal will be ~10x stronger for the upper band for the same amount of the product. The quantity of the upper product is much lower than the quantity of the lower band. This also explains higher variability of that product since minor changes in the presence are resulting in a variation of the signal. We think that some small RNAs dissociate in course of the assay and to them 15-20 nucleotides will be added. We added this to the text.

Minor comments

We improved the text according to the suggestions from Reviewer #2.

1. When the authors refer to the RNA exosome, there's some confusion about how the core exosome and its associated exonucleases and co-factors interact and function, which should be revised:
a. abstract: "the nuclear exosome Rrp6". Rrp6 is an exonuclease that associates with the core exosome.

We changed the sentence with "3'-5' exonuclease Rrp6".

My comment: It is changed in the abstract, but still wrong in the discussion and even in the rebuttal letter.

We fixed it now.

7. The results section, regarding centromeric repeats and silencing, second paragraph on page 7, could contain a clearer description. In the first sentence "we observed only a smaller reduction...", smaller than what? The second last sentence on the page states "... is only moderately reduced", there could also be a comparison to the *clr4*Δ strain, so that it is clear what "moderately" refers to. We included the data of *dcr1* deletion strain which shows a complete derepression of centromeric silencing. When compared to *dcr1* deletion, the loss of silencing in *cid14/16* deletion strain is small.

My comment: It is good to include the *dcr1* deletion for comparison. Regarding the first sentence, at least now it is clear that it is small in relation to wild-type cells, but I am not sure to what extent a reduction from 85% to 57 % can be considered small.

We included now the Ago1-bound small RNA classes from *dcr1* deletion cells (Fig. 2b, Marasovic et al., 2013). In comparison to the drop of centromeric small RNAs in *dcr1* deletion strain (0.1 %), reduction from 85% to 57% can be considered small.

Reviewer #3 (Remarks to the Author):

The authors have successfully addressed my concerns regarding genome-wide analysis of the activity of euchromatic siRNAs and *Cid14/16*-based degradation of Argonaute-associated sRNAs. I recommend publication.

We thank the reviewer for positive evaluation.

REVIEWERS' COMMENTS:

Reviewer #1 (Remarks to the Author):

One of the main findings of this paper is a suggestion that accumulation of siRNAs in Cid14 and Cid14/Cid16 knockout cells leads to silencing of RNA dependent RNA polymerase (Rdp1) which according to the Authors “protect the genome from uncontrolled RNAi”. The problem is that based on the RNA-seq data the Rdp1 mRNA levels does not change much in the Cid14 knockout (~25% reduction). In contrast based on Q-rtPCR the Authors claims that in Cid14 knockout Rdp1 is reduced 2 fold and 4 fold in the Cid14/Cid16 double mutant. Thus, the data are inconsistent. Why it is so? The inconsistency is not discussed in the manuscript at all .

Importantly, because of such inconsistency the Authors should analyze expression levels of Rdp1 more thoroughly by the northern blot for examples (Q rtPCR detect and quantify a small fragment of cognate mRNA only). Furthermore, if the levels of Rdp1 are indeed reduced in the mutant cells the mRNA stability should be analyzed in order to prove that the observed effect is posttranscriptional.

The other issues raised by me in the previous round of the review process are generally solved although I am not completely convinced that Rrp6 is responsible for the decay of tailed Argonaute bound siRNAs. There is no proof that adenylation or uridylation recruits Rrp6 in vivo. Thus, such overstatements should be removed from the manuscript.

Reviewer #1 (comments to authors)

One of the main findings of this paper is a suggestion that accumulation of siRNAs in Cid14 and Cid14/Cid16 knockout cells leads to silencing of RNA dependent RNA polymerase (Rdp1) which according to the Authors “protect the genome from uncontrolled RNAi”. The problem is that based on the RNA-seq data the Rdp1 mRNA levels does not change much in the Cid14 knockout (~25% reduction). In contrast based on Q-rtPCR the Authors claims that in Cid14 knockout Rdp1 is reduced 2 fold and 4 fold in the Cid14/Cid16 double mutant. Thus, the data are inconsistent. Why it is so? The inconsistency is not discussed in the manuscript at all . Importantly, because of such inconsistency the Authors should analyze expression levels of Rdp1 more thoroughly by the northern blot for examples (Q rtPCR detect and quantify a small fragment of cognate mRNA only). Furthermore, if the levels of Rdp1 are indeed reduced in the mutant cells the mRNA stability should be analyzed in order to prove that the observed effect is posttranscriptional.

In RNAseq data rdp1 mRNA level was reduced to 0.7 (1 replicate) and in qPCR to 0.5 (4 replicates). We discuss both experiments in the manuscript now. For me these data are not inconsistent as pointed by the reviewer, but reflect variation between methods and number of replicates. Using two independent methods we observe the same overall trend that rdp1 mRNA level is reduced. And, importantly, we show that over-expression of rdp1 mRNA leads to increased ectopic silencing in cid14 deletion cells, but not in wild type cells.

The other issues raised by me in the previous round of the review process are generally solved although I am not completely convinced that Rrp6 is responsible for the decay of tailed Argonaute bound siRNAs. There is no proof that adenylation or uridylation recruits Rrp6 in vivo. Thus, such overstatements should be removed from the manuscript.

We removed overstatements saying that adenylation or uridylation recruits Rrp6 in vivo. Our in vitro data show that adenylation and uridylation are required for degradation of Argonaute-bound small RNAs by Rrp6. In vivo we observed reduced small RNA degradation in cid14 deletion cells, which shows that Cid14 is required for degradation of Argonaute-bound small RNAs. Cid14 is a component of the TRAMP complex that recruits Rrp6. And we have already observed that Argonaute-bound small RNAs accumulate in Rrp6 deletion cells in our previous work (Marasovic et al, 2013). All this points out that Rrp6 degrades Argonaute-bound small RNAs in vivo as well.